# ExPO: Unlocking Hard Reasoning with Self-Explanation-Guided Reinforcement Learning

**Ruiyang Zhou**♠
University of Texas at Austin
ruiyang.zhou@utexas.edu

**Shuozhe Li**♠
University of Texas at Austin
shuozhe.li@utexas.edu

**Amy Zhang**
University of Texas at Austin

**Liu Leqi**♠
University of Texas at Austin
leqiliu@utexas.edu*

## Abstract

Self-improvement via RL often fails on complex reasoning tasks because GRPO-style post-training methods rely on the model's initial ability to generate positive samples. Without guided exploration, these approaches merely reinforce what the model already knows (distribution-sharpening) rather than enabling the model to solve problems where it initially generates no correct solutions. To unlock reasoning ability in such settings, the model must explore new reasoning trajectories beyond its current output distribution. Such exploration requires access to sufficiently good positive samples to guide the learning.

While expert demonstrations seem like a natural solution, we find that they are often ineffective in RL post-training. Instead, we identify two key properties of effective positive samples: they should (1) be likely under the current policy, and (2) increase the model's likelihood of predicting the correct answer. Based on these insights, we propose **Self-Explanation Policy Optimization (ExPO)**—a simple and modular framework that generates such samples by conditioning on the ground-truth answer. It can be integrated with popular RL training methods like GRPO and DPO. ExPO enables efficient exploration and guides the model to produce reasoning trajectories more aligned with its policy than expert-written CoTs, while ensuring higher quality than its own (incorrect) samples. Experiments show that ExPO improves both learning efficiency and final performance on reasoning benchmarks, surpassing expert-demonstration-based methods in challenging settings such as MATH level-5, where the model initially struggles the most. Code available in https://github.com/HumainLab/ExPO_rl_reasoning_by_explanation.

## 1 Introduction

The development of reinforcement learning (RL)-style post-training methods has been a pivotal factor in the advances of large language models on complex reasoning tasks. RL training optimizes model outputs using reward signals derived from human preferences, model comparisons, or automated verifiers [16, 25, 27, 24]. RL-style post-training encompasses a broad family of algorithms, including reward-maximizing reinforcement learning (e.g., GRPO) [20, 1], contrastive preference optimization methods [23, 38, 34, 39], and supervised fine-tuning on expert demonstrations or high-quality self-generated samples [8]. Despite differences in algorithmic formulation, all these methods share a central mechanism: increasing the likelihood of generating preferred/positive responses and reducing that of dispreferred/negative ones.

---

*♠: Leading Contributors (Equal Contribution).

39th Conference on Neural Information Processing Systems (NeurIPS 2025).

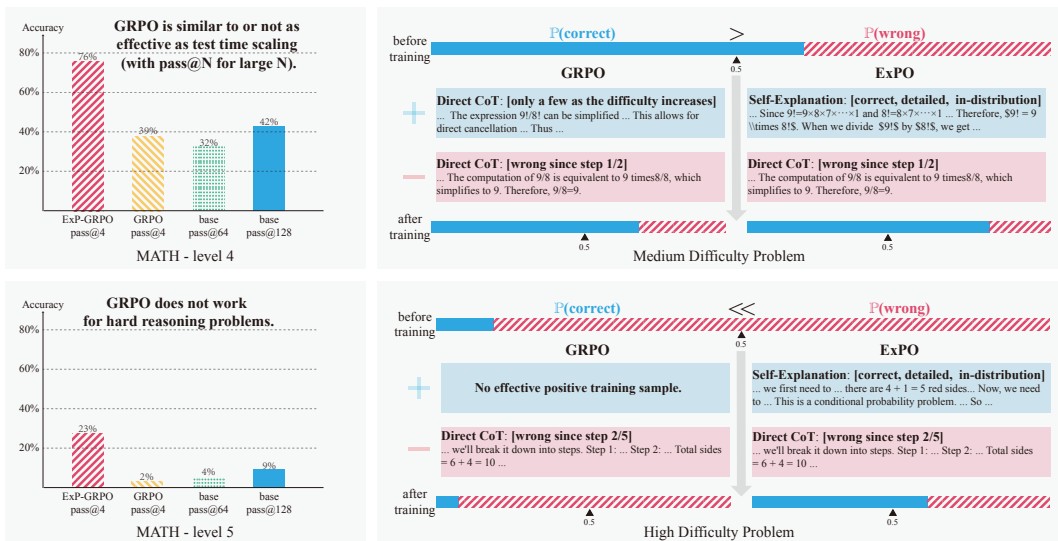

Figure 1: Illustration of the problem and our proposed solution ExPO. On the left, models (base model is Qwen2.5-3B-Instruct) performance on the MATH dataset highlights the issue with GRPO-style methods: they primarily strengthen the model's existing capabilities rather than enabling new ones. On the right, we present the positive and negative samples of both GRPO and our proposed ExPO method for MATH level-4 (top) and level-5 (bottom). ExPO is more effective than GRPO in guiding the model to learn for hard reasoning tasks.

While negative samples are typically abundant during RL post-training, positive samples are scarce, especially in hard reasoning tasks. Thus, a key challenge remains unresolved: *how to obtain effective positive training samples in settings where the model's initial success rate is low*? In practice, positive samples are either drawn from expert demonstrations, which are costly to obtain for complex domains (e.g., MATH level-5 [12]), or self-generated by the model. Yet for hard problems, the model often fails to produce any correct answers initially, leading to a pathological regime in GRPO-style training: the advantage term vanishes, the KL term dominates, and the model regresses—an effect sometimes described as "unlearning" [11, 43].

Existing work attempts to circumvent this by discarding training examples with all incorrect responses [37, 35], but this merely sidesteps the core issue. Fundamentally, GRPO-style methods assume high-quality samples are likely to be sampled under the model's current policy, but this assumption breaks down in high-difficulty regimes (Figure 1). In other words, these methods exhibit a **distribution-sharpening bias**: they sharpen the model's output distribution around high-probability correct responses, but struggle to guide learning on tasks where correct answers are unlikely under the current model. This limitation reflects a broader issue noted in recent work [40]: current RL-style post-training often reinforces existing capabilities rather than fostering fundamentally stronger reasoning abilities beyond those of the base model.

This motivates a fundamental question:

> **How can we effectively guide learning for challenging reasoning tasks in RL-style post-training, when positive samples are scarce?**

Rather than focusing on suppressing negative samples that offer little benefit when the model lacks any useful priors, we argue that progress hinges on identifying and synthesizing *effective positive samples*. This paper therefore investigates two core questions:

1. **How can we obtain positive samples when the base model's initial success rate is low?**
2. **What properties must such samples possess to provide strong learning signals in RL-style post-training?**

**Our Approach: ExPO.** To answer these questions, we begin by identifying two essential properties of ideal positive training samples:

- **In-distribution**: The sample should have a high probability under the current policy to guide learning effectively.
- **Positive learning signal**: The sample should increase the likelihood of the correct answer compared to the model's current CoT.

These criteria inform our proposed method, **ExPO** (Self-**Ex**planation **P**olicy **O**ptimization), a modular framework for generating and integrating positive samples via self-explanation. Specifically, ExPO conditions the model on both the question and the final answer to generate plausible reasoning chains, which are then used as positive samples for RL-style training.

Our key insight is that conditioning on the ground-truth answer helps the model produce in-distribution reasoning traces that are more aligned with its current policy than expert-written CoTs, while providing better guidance than its own incorrect completions. These self-explanations are more elaborate and contain more correct steps than standard CoTs, serving as a form of natural *process supervision*—as their relatively small deviation from standard CoTs helps the model identify where and how to adjust its reasoning. Because they are more likely under the current policy, these self-explanations guide the model to explore more effectively than expert CoTs—enabling the model to learn in settings previously dismissed as unlearnable.

ExPO is broadly applicable and can be instantiated in both contrastive (e.g., DPO) and reward-based (e.g., GRPO) frameworks. Across both domains, we demonstrate that ExPO improves sample efficiency, accelerates learning, and significantly boosts performance—especially on difficult questions where prior methods fail. Notably, ExPO not only removes the need for expert CoTs, but also *outperforms* approaches that rely on them.

## 2   Related work

**RL methods for training LLM reasoning ability.** Reinforcement Learning methods are increasingly showing promise in training for complex reasoning tasks. The simple but effective Direct Preference Optimization (DPO) algorithm [23] is used for reasoning and other human preference alignment tasks on a pair of positive and negative generations. There are variants to better apply DPO on reasoning tasks, for example, the iterative DPO method [21, 29] iteratively applies DPO to make it online; the self-training DPO method [31] iterates between SFT and DPO. On the other hand, the Group Relative Policy Optimization (GRPO) method [25] uses only the outcome verifier to train strong reasoning models. Without the need for process supervision signal (reasoning chain-of-thought), it is possible to train on a large scale of tasks. We incorporate our method ExPO with DPO and GRPO to further improve their training performance on challenging reasoning tasks.

**RL methods for self-improvement without expert-labeled CoT.** Given the challenge of acquiring expert-labeled CoT for complex reasoning tasks, self-improvement through reinforcement learning has become a key area of research. Current approaches include reinforcement learning from AI feedback [18, 9, 33], which uses a powerful "judge" model to evaluate responses; methods with iterative self-correction and self-refinement, where a model refines its own generation [2, 22]; and the use of self-generated training data [41, 7]. Our work contributes to this third category, where the model generates the reasoning steps as signals for its own training.

One important paper in this domain is STaR [41], which proposes to use self-generated chain-of-thought during finetuning on reasoning tasks. In the absence of expert-labeled CoT, they prompt the model to regenerate rationales for incorrect predictions given the correct answer. While STaR is limited to SFT training and direct prompting, subsequent work proposes various methods for better self-training with RL. Their methods include using model rules, implicit signals, or external verifiers [13–15, 28, 42]. However, while these approaches demonstrate practical success, they do not explain why STaR-style methods are so effective. Our paper proposes a more general theoretical analysis on why STaR-style methods can work and sometimes even surpass methods that use expert-labeled CoT, particularly under the RL framework.

## 3   Characteristics on the Ideal Positive Training Sample

We focus on settings where the model struggles to generate positive samples on its own—though it can still produce negative ones (e.g., incorrect self-generated responses), as commonly seen in

hard reasoning tasks. In such cases, effective positive training samples are essential to guide the model's exploration and reduce the sample complexity of learning. To tackle this challenge, we first analyze what makes a positive training sample effective in the context of policy/preference optimization for reasoning tasks (Section 3.2, 3.3). We then demonstrate how our proposed method for generating positive samples satisfies these ideal properties (Section 4). Finally, we introduce ExPO (self-Explanation-based Policy Optimization), along with instantiations using different base $^\star$PO algorithms such as DPO and GRPO (Section 5).

## 3.1 Problem Setup

When training a large reasoning model, our (implicit) end goal is the following:

$$\max_\theta \mathbb{E}_{(q,a^\star)\sim\mathcal{D},(c,a)\sim\pi_\theta(\cdot|q)} \left[ r(q,c,a,a^\star) \right], \tag{1}$$

where $\mathcal{D}$ is a distribution of question $q$ and ground-truth answer $a^\star$ pairs. The model $\pi_\theta$ generates a pair of CoT $c$ and answer $a$, and the verifiable reward is often given by whether the answer is correct, i.e., $r(q,c,a,a^\star) = \mathbb{I}\{a = a^\star\}$. Though many algorithms (e.g., $^\star$PO algorithms like GRPO, DPO, PPO, etc.) have been proposed to train reasoning models with various learning objectives, implicitly, our end goal is Eq. (1) as indicated by how we have evaluated the performance of the trained model $\pi_\theta$: we evaluate them directly on their correctness on reasoning tasks instead of evaluating them on their learning objectives. For simplicity, we denote the objective in Eq. (1) as $\boldsymbol{J}(\theta)$.

To understand the effect of data on the training process across different algorithms, we consider two complementary types of analysis: Policy improvement[26], where we use gradient-based analysis to examine how different samples contribute to optimizing the overarching objective Eq. (1); and Probability shifts [5], where we analyze directly how the policy $\pi_\theta$ changes in response to different types of training samples (and that these probability shifts are closely tied to policy improvement).

Drawing insights from both analyses, we identify two key properties that characterize ideal positive training samples: (1) They must be **in-distribution** with respect to the current policy; and (2) They must be better than negative samples in achieving the task at hand (which we will provide a more precise definition for) and thus provide **positive learning signal**.

## 3.2 Property 1: In-distribution

Consider a generic gradient $\boldsymbol{g}(\bar{q},\bar{c},\bar{a})$ obtained with a sample (question, CoT, answer) pair $(\bar{q},\bar{c},\bar{a})$. For now, we keep the gradient definition abstract: it can be derived from any policy/preference optimization algorithm, and the sample used for its computation can be obtained on- or off-policy . In this setting, after taking a gradient ascent[2] step with learning rate $\alpha > 0$, we have $\theta_{t+1} = \theta_t + \alpha \boldsymbol{g}(\bar{q},\bar{c},\bar{a})$, and approximate the change in the objective (1) by a first-order Taylor expansion:

$$\Delta\boldsymbol{J} = \boldsymbol{J}(\theta_{t+1}) - \boldsymbol{J}(\theta_t) \approx \alpha \nabla_\theta \boldsymbol{J}(\theta)^\top \boldsymbol{g}(\bar{q},\bar{c},\bar{a}). \tag{2}$$

The key to understand the effect of training data $(\bar{q},\bar{c},\bar{a})$ on the policy improvement $\Delta\boldsymbol{J}$ is thus to identify the alignment between the true gradient $\nabla_\theta \boldsymbol{J}(\theta)$ and the sample gradient $\boldsymbol{g}(\bar{q},\bar{c},\bar{a})$ used in training. Given the definition in (1), we have the true gradient to be

$$\nabla_\theta J(\theta) = \mathbb{E}_{(q,a^\star)\sim\mathcal{D},(c,a)\sim\pi_\theta(\cdot|q)} \left[ \nabla_\theta \log \pi_\theta(c,a|q) r(q,c,a,a^\star) \right], \tag{3}$$

As discussed in Shao et al. [25] Appendix A, given the training data $(\bar{q},\bar{c},\bar{a})$, the gradient of $^\star$PO algorithms often takes the form:

$$\boldsymbol{g}(\bar{q},\bar{c},\bar{a}) = w \nabla_\theta \log \pi_\theta(\bar{c},\bar{a}|\bar{q}), \tag{4}$$

for some weighting $w \in \mathbb{R}$. The weighting depends on the *relative quality* of the data. When a sample is considered "positive"—that is, it performs better than other samples in the batch (e.g., it achieves a higher objective value or is correct)—the weighting is positive ($w > 0$). Conversely, samples that perform worse has the weighting to be $w < 0$.[3]

---

[2]Depending on the exact $^\star$PO algorithm (minimizing or maximizing the learning objective), it can be either a descent or ascent step.

[3]For certain algorithms (e.g., PPO), the weighting is applied per-token, i.e., the weighting is different for each of the summand in $\nabla_\theta \log \pi_\theta(\bar{c},\bar{a}|\bar{q}) = \sum_t \nabla_\theta \log \pi_\theta(\bar{c}_t|\bar{q},\bar{c}_{<t}) + \nabla_\theta \log \pi_\theta(\bar{a}|\bar{q},\bar{c})$. For illustration simplicity, we keep a fixed weighting for each token in a (response) trajectory, but the general properties we illustrate below apply to settings with token-dependent weighting.

By injecting the $\nabla_\theta J(\theta)$ and $\boldsymbol{g}(\bar{q}, \bar{c}, \bar{a})$ into the objective3.2 and transform the expectation into summation, we argue that the magnitude of the policy improvement $\Delta \boldsymbol{J}$ (and the alignment between the true gradient $\nabla_\theta \boldsymbol{J}(\theta)$ and the actual used gradient $\boldsymbol{g}(\bar{c}, \bar{a}, \bar{q})$) is dominated by the term (details are in Appendix)

$$w\pi_\theta(c', a'|q')\|\nabla_\theta \log \pi_\theta(c', a'|q')\|^2 r(q', c', a', a^\star), \tag{5}$$

Thus, the training data $(\bar{q}, \bar{c}, \bar{a})$ will only be effective in improving the policy if $\pi_\theta(\bar{c}, \bar{a}|\bar{q})$ **is large**, i.e., the training sample $(\bar{q}, \bar{c}, \bar{a})$ is "**in-distribution**" with respect to policy $\pi_\theta$.

We now formalize the first ideal property of a positive training sample: being in-distribution under the current policy.

**Property 1** (**In-distribution**). *A sample $(\bar{c}, \bar{a})$ is considered more in-distribution relative to another sample $(c, a)$ for a given question $q$ if its probability under the current policy $\pi_\theta$ is higher, i.e.,*

$$\pi_\theta(\bar{c}, \bar{a}|q) > \pi_\theta(c, a|q).$$

This is a relative criterion, not an absolute one, reflecting the fact that in practice, when determining which samples in a given batch are more in-distribution, we compare how likely each sample is under the current policy. Samples with high probability and high reward can contribute more effectively to policy improvement. In practice, we want the training sample $(\bar{q}, \bar{c}, \bar{a})$ to have $\pi_\theta(\bar{c}, \bar{a}|\bar{q})$ sufficiently far from zero, and not too far from $\max_{c,a} \pi_\theta(c, a|\bar{q})$. This ensures that the sample is in-distribution enough to meaningfully contribute to policy improvement.

### 3.3 Property 2: Positive Learning Signal

In the above, we already see the importance of having in-distribution data: gradients for samples $(\bar{q}, \bar{c}, \bar{a})$ (with positive rewards) where $\pi_\theta(\bar{c}, \bar{a}|\bar{q}) \gg 0$ enable more effective policy improvement. In the following analysis, we examine which in-distribution samples should be given positive rewards and receive positive learning signals to increase their probability.

Consider a batch containing two training samples for the same question: $\{(q, c_1, a_1), (q, c_2, a_2)\}$. When $a_1$ is correct and $a_2$ is not, it is clear that the correct one should be preferred, i.e., $(q, c_1, a_1) \succ (q, c_2, a_2)$, and thus receive the positive reward to increase its probability. However, in settings where both $a_1$ and $a_2$ are incorrect (as is often the case in model training for challenging reasoning tasks), which sample should be treated as the positive/preferred one to increase the probability?

We argue that the notion of preference between responses in general should be defined in terms of their ability to increase the likelihood of the correct answer $a^\star$. Specifically, we propose the following criterion:

**Property 2** (Positive learning signal). *A sample $(q, c_1, a_1)$ has a positive learning signal compared to $(q, c_2, a_2)$, and thus should be preferred, i.e., $(q, c_1, a_1) \succ (q, c_2, a_2)$, if and only if*

$$\pi_\theta(a^\star|q, c_1) > \pi_\theta(a^\star|q, c_2).$$

We note that this property is also defined in a relative sense: it allows us to compare two chain-of-thoughts, but does not assign absolute quality to any individual sample. In the specific case of a batch containing two samples, if $(q, c_1, a_1)$ carries a positive learning signal relative to $(q, c_2, a_2)$, then training algorithms should assign the weighting $w$ for the gradient $\boldsymbol{g}(q, c_1, a_1)$ to be positive (hence, positive gradient), and the weighting $w$ for the gradient $\boldsymbol{g}(q, c_2, a_2)$ to be negative (4).

Why should we increase the probability of $(q, c_1, a_1)$ and decrease that of $(q, c_2, a_2)$, if $(q, c_1, a_1)$ possess a positive learning signal as defined in Property 2? For a fixed question and ground truth answer pair $(q, a^\star)$, consider our objective defined in (1):

$$\boldsymbol{J}(\theta; q, a^\star) = \sum_{a,c} \pi_\theta(c|q)\,\pi_\theta(a|q, c)\,\mathbb{I}\{a = a^\star\} = \sum_c \pi_\theta(c|q)\,\pi_\theta(a^\star|q, c). \tag{6}$$

This objective reflects the total probability mass assigned to generating the correct answer $a^\star$ through all reasoning paths $c$. Now, if $\pi_\theta(a^\star|q, c_1) > \pi_\theta(a^\star|q, c_2)$, then increasing $\pi_\theta(c_1|q)$ and decreasing $\pi_\theta(c_2|q)$ would increase the overall objective $\boldsymbol{J}(\theta; q, a^\star)$. Therefore, adjusting the probabilities in this way directly improves the model's ability to generate the correct answer—justifying the preference defined in the property. In Appendix, we provide a detailed analysis of how increasing or decreasing probabilities under a softmax policy influences others (based on their relative value), showing that choosing the wrong samples to increase/decrease probability can lead to unintended effects.

# 4 Positive Training Sample Generation through Self-Explanation

As discussed, the ideal positive training samples should satisfy two properties: **Property 1** (in-distribution), which ensures the samples to have relatively high probability under the current training policy $\pi_\theta$, and **Property 2** (positive learning signal), which specifies that within a batch, the sample whose CoT makes the ground truth answer more probable—relative to the CoTs of other samples—should receive a positive gradient, i.e., its probability should be up-weighed.

In scenarios where self-generated responses (CoT, answer pairs) perform poorly (e.g., on challenging tasks where the model has not yet learned to perform well), a surprisingly simple method can yield positive samples that satisfy the **in-distribution** and **positive learning signal** properties—generate a **self-explanation conditioned on the correct answer**:

$$\tilde{\boldsymbol{c}} \sim \pi_\theta(\cot | q, a^\star), \tag{7}$$

and use $(q, \tilde{c}, a^\star)$ as a positive sample. This idea resembles the sampling strategy proposed in STAR [41][4]. It is worth noting that the criteria for ideal positive training samples (**Property 1, 2**) are more general than this specific sampling approach. As long as a sample is probable under the training policy and leads towards the correct answer more likely on average than the self-generated CoT, then the sample is ideal for providing positive learning signals. For example, beyond generating the sample conditioned on the answer, one can generate it by conditioning on a partial expert CoT or with additional hints.

## 4.1 Self-Explanation: Natural Guidance for RL Post-training

We begin by showing an example of self-explanation and provide some intuition on why self-explanations are suitable positive training samples in RL post-training. Example 1 shows a self-explanation for a hard math problem. For this problem, conditioning the model on the correct answer enables it to successfully identify the pivotal step and guide the reasoning process correctly. Compared to the model's standard CoT, the self-explanation often contains more correct reasoning steps. Besides, the self-explanation's phrasing is more consistent with the model's own language, offering a clearer contrast to its incorrect responses.

Intuitively, self-explanation reduces the task difficulty by providing the correct answer as a known condition, shifting the challenge from open-ended problem-solving to generating conditional explanations. Because of this, the self-generated explanation has better quality while having a relatively small deviation from the incorrect standard CoT. These properties seem well-suited for RL-style training, which, in our understanding, is more geared toward shaping the existing behavior of the base model rather than instilling entirely new knowledge. Therefore, self-explanation helps the model identify where and how to adjust its reasoning, functioning as an effective and natural form of process supervision.

---

**Example 1: Standard CoT vs self-explanation for a question from MATH**

**Question**: A Conditional probability question... Given that the side you see is red, what is the probability that the other side is red?

**Standard CoT**: ✓[Correct step 1] Step 1: Count... ✗[Redundant step 2] Step 2: Determine... ✗[Wrong computation in step 3] Step 3: Calculate...

**Self-Explanation**: ✓[Correct step 1] To solve this problem... ✓[Correct step 2] Now, we need to find... ✓[Explain the answer again] The answer is...

---

In fact, if the base model still fails to generate better reasoning traces with the guidance given by the correct answer, the problem is likely too difficult for the RL post-training approach. Compared to supervised fine-tuning (SFT), RL training is not as efficient at teaching new knowledge to the model. The objective of SFT is to maximize the probability of the target response, so that the knowledge in the response can be memorized. In contrast, the RL post-training objective is to make the positive/chosen response more likely than the negative/rejected one, which provides a weaker learning signal for memorizing complex, response-relevant new knowledge. Besides, the KL term in the RL post-training objective suggests that the trained model always stays close to the base model.

---

[4]We arrived at this sampling strategy independently, and only later recognized its resemblance to prior work.

Therefore, if the model is unable to articulate improved reasoning even when the correct answer is known, it is unlikely that reward-driven learning alone will yield meaningful gains in reasoning capability—at least not in an efficient manner.

## 4.2 Comparing Self-Explanation with Expert CoT and Standard CoT

To more formally understand why self-explanations (7) serve as ideal positive samples, we compare them—both empirically and analytically—with two other types of data: the expert CoT $c_E$ and the original self-generated CoT $c$. Self-generated explanations satisfy both ideal properties (Table 1). In contrast, expert CoT lacks Property 1 and, empirically, provides less effective guidance for model learning (Section 6). Compared to standard CoT, self-explanations provide positive learning signals. We elaborate on these findings below.

| Method | In-distribution | Positive Learning Signal |
|---|---|---|
| Ours (self-explanation) | ✓ | ✓ |
| Expert CoT | ✗ | ✓ |
| Standard CoT | ✓ | ✗ |

Table 1: Comparison of different training data with respect to the ideal properties. Our ExPO method verifies both.

First, regarding the **in-distribution** Property 1, we observe that, compared to the expert CoT $c_E$, the self-generated explanation $\tilde{c}$ is more likely to be generated under the current policy and thus more in-distribution for training. As shown in Figure 2, the self-generated explanation has a significantly lower negative log-likelihood than the expert CoT. Surprisingly—at least empirically—training with expert CoT can sometimes result in performance worse than the performance of models trained using self-generated explanations (as shown in Section 6). This observation aligns with our gradient-based analysis in Section 3.1. In general, the distribution of $\tilde{c} \sim \pi(\cdot|q, a^\star)$ is close to the original CoT distribution $c \sim \pi(\cdot|q)$, since the prompts used to generate these two distributions differ by only a small number of tokens, which describe the ground-truth answer $a^\star$.

Second, for the **positive learning signal** Property 2, as illustrated in Example 1 above—and consistently observed across the self-explanation data we have generated—self-generated explanations tend to be more detailed and contain more correct reasoning steps than the original self-generated CoT. This observation is further supported empirically by prompting GPT-4o to evaluate the relative quality of $\tilde{c}$ and $c$—GPT-4o consistently rated $\tilde{c}$ significantly higher in quality (Figure 2).

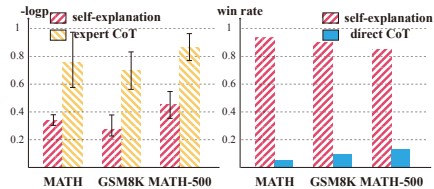

This intuition is also formalized in the following lemma:

**Lemma 1.** *On average, the self-generated explanation is more likely to lead to the ground-truth answer than the original self-generated chain-of-thought. That is,*

$$\mathbb{E}_{\tilde{c} \sim \pi_\theta(\tilde{c}|q, a^\star)} \left[ \pi_\theta(a^\star|q, \tilde{c}) \right] \geq \mathbb{E}_{c \sim \pi_\theta(c|q)} \left[ \pi_\theta(a^\star|q, c) \right].$$

Figure 2: Left: Negative log-likelihood of the self-explanation $\tilde{c}$ and expert CoT. Right: Winrate labelled by GPT-4o in terms the number of correct steps of the self-explanation $\tilde{c}$ and self-generated CoT $c$. Both on the test split of each dataset using Qwen2.5 3B-Instruct.

To summarize, the self-generated explanation $\tilde{c}$ outperforms the expert CoT $c_E$ with respect to the in-distribution property (though it is weaker in terms of the positive learning signal property), while it surpasses the self-generated CoT $c$ in terms of the positive learning signal property but not the in-distribution one. Thus, our proposed positive sample—the self-generated explanation $\tilde{c}$—achieves a balance between these two desirable properties. In the following, we describe how to leverage these generated positive samples within RL-type training for reasoning tasks.

# 5 ExPO and its instantiations on DPO and GRPO

Building on our method for generating in-distribution positive samples (Section 4) and the theoretical foundations established in Section 3.1, we instantiate our approach within two widely used policy/preference optimization frameworks: Direct Preference Optimization (DPO) and Group Relative Policy Optimization (GRPO). This leads to two practical algorithms **ExP-DPO** and **ExP-GRPO** that integrate explanation-based data augmentation into the learning pipeline.

## 5.1 ExP-DPO

We design and evaluate two variants of ExP-DPO: an offline and an iteratively online version. In the offline DPO setting, all explanations $\tilde{c}$ and responses $c_-$ are generated by the initial policy at the beginning of the training, i.e. $\tilde{c}_+ \sim \pi_{\text{ref}}(\cdot|q, a^\star)$ and $c_- \sim \pi_{\text{ref}}(\cdot|q)$. In the online DPO setting, $\tilde{c}$ is iteratively generated from the up-to-date policy.

**Observation 1.** In the offline setting, we implement ExP-DPO using either the expert-provided CoT $c_E$ or the model-generated explanation $\tilde{c}$ as the winning response (see Section 6). However, training with $c_E$ often results in deceptively low loss values, yet fails to yield strong downstream performance. This occurs because $c_E$ typically has a much lower log-probability compared to the self-generated CoT (the losing response). As a result, during training, even a minor increase in the expert's log-probability can suffice to classify it as the correct answer without meaningfully increasing the expert CoT's log-probability relative to that of the self-generated one.

**Observation 2.** While being effective at the beginning, the offline scheme eventually suffers from distributional drift: as the policy $\pi_\theta$ updates, the fixed explanation $\tilde{c}$ becomes increasingly out-of-distribution, violating the in-distribution criterion outlined in Property 1.

These observations correspond with our theoretical analysis: out-of-distribution positive samples can yield ineffective or even detrimental learning signals. To address this, we introduce the iteratively online ExP-DPO, in which a fresh explanation $\tilde{c}_+$ is regenerated after updating $\pi_\theta^i$ over a large enough batch of data, i.e. $\tilde{c}_+ \sim \pi_\theta^i(\cdot|q, a^\star)$. This formulation ensures that the winning sample remains in-distribution of the evolving model, thereby providing a consistently strong learning signal.

## 5.2 ExP-GRPO

The Group Relative Policy Optimization (GRPO) method derives its learning signal by sampling responses from the policy model itself. However, this approach has a critical limitation: when all sampled responses are incorrect, the model receives no effective gradient signal. In such cases, training degenerates into minimizing the KL divergence term alone, which fails to guide the model toward better performance. This issue is particularly acute when the sampled questions are too difficult for the model to generate valid intermediate reasoning steps. Prior works have circumvented this problem by removing the KL divergence term or excluding such challenging examples from training altogether [37, 19, 32], but they stop short of addressing the fundamental question—how can a model learn to solve problems it currently fails at? Our method provides a principled mechanism to overcome this limitation via the generated self-explanation $\tilde{c}$ and enables the model to explore and learn even when initial sampled responses are incorrect, thereby unlocking the potential of the model to improve on previously unsolvable instances.

In our design, we address such issues by introducing an ExP-SFT term with a scaling coefficient $\beta$. We fix $\beta = 0.04$ in all reported experiments and an ablation study to justify this choice across multiple $\beta$ values on the full MATH training set is in Appendix. Specifically, we use the initially generated $\tilde{c}$ as the CoT that leads to the correct answer. Then, we reconstruct the GRPO objective as:

$$\mathcal{J}_{\text{GRPO}}(\theta) = \mathbb{E}_{\substack{(q,a^\star)\sim\mathcal{D}, \\ \{o_i\}_{i=1}^G \sim \pi_{\theta_{\text{old}}}(\cdot|q) \\ \tilde{c}\sim\pi_\theta(\cdot|q,a^\star)}} \left[ \frac{1}{G}\sum_{i=1}^G \frac{1}{|o_i|}\sum_{t=1}^{|o_i|} \min\left(r_{i,t}(\theta)\hat{A}_{i,t}, \text{clip}(r_{i,t}(\theta), 1-\varepsilon, 1+\varepsilon)\hat{A}_{i,t}\right) + \boxed{\beta\log\pi_\theta(\tilde{c}, a^\star|q)} \right]$$

**Key Observations.** In Section 6, we empirically demonstrate that the ExP-SFT term directly addresses the core challenge of missing learning signals in difficult reasoning settings—where correct responses are exceedingly rare under the initial policy. By injecting in-distribution, task-relevant supervision through $\tilde{c}$, our method activates the trial-and-error learning loop that is otherwise stalled, enabling the model to make meaningful progress even on instances previously considered unlearnable. Moreover, we show that the addition of the ExP-SFT term significantly enhances sample efficiency and allows the policy to reach a higher performance ceiling than baselines without such augmentation. Further analysis reveals that the majority of the performance gains originate from the harder questions in the test set. This suggests that the reasoning capabilities acquired via the ExP-SFT term not only improve training efficiency but also improve models' reasoning capability on more challenging, unseen instances.

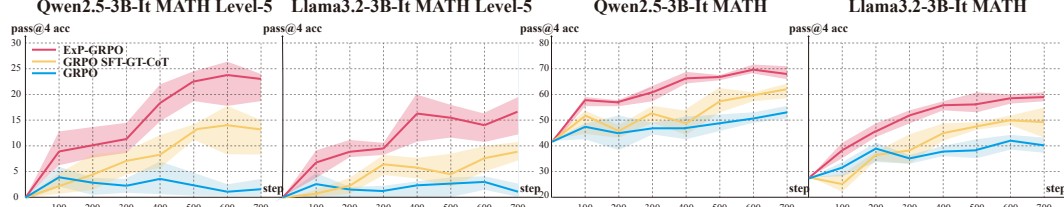

Figure 3: Accuracy on the **level-5** questions from MATH test set (left 1, 2) and on the whole test set (right 3, 4) for Qwen2.5-3B-Instruct and LLaMA-3.2-3B-Instruct across global training steps. **ExP-GRPO** consistently outperforms both **GRPO** and **GRPO SFT-GT-CoT**, the latter uses supervised fine-tuning on expert CoT $c_E$. The results show that **ExP-GRPO** provides more effective and generalizable learning signals, leading to improved sample efficiency and higher overall performance.

We can interpret the ExP-SFT term as a form of natural process supervision—it increases the likelihood of generating self-explanations $\tilde{c}$ while decreasing that of the self-generated CoTs $c$. The relatively small deviation between these two reasoning traces helps the model pinpoint where and how to adjust. This mechanism is particularly valuable for challenging tasks, where the model may initially fail. In such cases, it is crucial to provide effective positive learning signals that introduce new reasoning traces or knowledge. Compared to GRPO, ExP-SFT provides these signals and guides the model in the right direction, leading to greater sample efficiency and performance ceiling.

## 6 Experiments

We present empirical results demonstrating the effectiveness of ExPO by providing strong on-policy learning signals. We begin by evaluating our method in the context of ExP-DPO. The detailed experimental results and analysis are in Appendix D.3. We then turn to a more challenging scenario: GRPO [25] training when the initial policy struggles to generate meaningful responses and fails to produce informative learning signals. In this regime, our method ExP-GRPO is especially valuable: it jump-starts the learning process by guiding the policy early on, thereby igniting the trial-and-error loop that is critical for sustained improvements in reasoning.

**Models and training settings.** We conduct preference optimization experiments using two families of models: LLaMA-3.2 [6] and Qwen-2.5 [36]. Training and evaluation are performed on two widely used mathematical reasoning benchmarks: MATH [12] and GSM8K [3]. The self-explanation of ExP-GRPO training is published [5]. Other experimental details are in Appendix D.1.

### 6.1 Results on ExP-GRPO

To create scenarios where learning signals are hard to obtain, we picked the difficulty level-5 questions from the MATH dataset. In GRPO training, without extra guidance, these questions are hard for the initial policy to obtain any effective learning signal. In addition, obtaining expert CoT label for hard questions like these are expensive. However, with the help of learning signals from ExPO, the initial policy acquired effective learning signals and soon is able to further improve its performance via online sampling as shown in Figure 3. These results indicate our method's effectiveness on providing helpful learning signals on reasoning training set where learning signals are rare.

Further more, in training settings, using the whole MATH training set, that feature a mixture of easy and hard questions—each associated with differing availability of learning signals—we find that augmenting GRPO with the ExP-SFT term markedly improves both sample efficiency and the final performance ceiling, as shown in Figure 3 (see Appendix for performance on more datasets). This improvement stems from ExP-SFT's ability to consistently inject informative, on-policy learning signals even in regimes where standard policy samples fail to provide meaningful supervision. To better understand the source of these gains, we conduct a breakdown analysis by question difficulty. As summarized in Table 2, we observe that the majority of performance improvements arise from the hardest subset of questions in the test set. This suggests that the reasoning capabilities acquired

[5]https://huggingface.co/datasets/humainlab/MATH-self-explanation

|                        | Level 1 | Level 2 | Level 3 | Level 4 | Level 5 |
| ---------------------- | ------- | ------- | ------- | ------- | ------- |
| # Test Samples         | 437     | 894     | 1131    | 1214    | 1324    |
| **ExP-GRPO pass@4**    | **96%** | **91%** | **86%** | **76%**↑ | **23%**↑ |
| GRPO SFT-GT-CoT pass@4 | 95%     | 89%     | 83%     | 65%     | 12%     |
| GRPO pass@4            | 91%     | 84%     | 77%     | 39%     | 2%      |
| Base pass@64           | 97%     | 88%     | 75%     | 32%     | 4%      |
| Base pass@128          | 97%     | 93%     | 80%     | 42%     | 9%      |

Table 2: Accuracy of different methods (base model is Qwen2.5-3B-Instruct) across difficulty levels on the MATH test set. While all methods perform comparably on easier questions, the performance gap widens dramatically on harder levels. Especially on level-4 and level-5 questions, **ExP-GRPO** yields substantial gains while the standard GRPO fails to learn. Moreover, when evaluated under large pass@k settings—commonly used to reveal the full reasoning capacity of LLMs—**ExP-GRPO** exhibits a substantially broader coverage on difficult questions, effectively harnessing and expanding the model's latent problem-solving ability beyond the reach of conventional RL methods.

via the ExP-SFT signal not only facilitate more effective policy learning during training, but also generalize robustly to challenging, previously unsolvable instances at test time.

# 7 Discussion and Future Work

**Discussion of broader application.** While our experiments focus on math reasoning, ExPO's core idea of bootstrapping learning with self-generated, in-distribution samples conditioned on verifiable outcomes applies broadly. For example, in code generation (e.g., Codeforces), correct outputs can be verified via test cases. Given the desired output, the model can be prompted to generate rationale (e.g., pseudocode), enabling ExPO to provide effective training signals. In common-sense reasoning tasks where answers are deterministic, ExPO can generate the multi-step reasoning required to reach the correct answer. More generally, any reasoning task with verifiable rewards (such as physics or scientific QA) can benefit from this approach. Ultimately, ExPO demonstrates that a model can effectively teach itself without expert-labeled chain-of-thought, paving the way for enhancing language models' complex reasoning ability.

**Discussoin of the exploratory role of the advantage-weighted objective.** ExPO method uses the relative quality as learning signals of the self-generated explanation and direct chain-of-thought. While Lemma 2 guarantees that the better-but-may-be-imperfect self-explanations often provide partial or heuristic reasoning paths that still guides the model to learn, we want to further discuss why the imperfect CoTs will not make the model collapse. On the one hand, our method does not collapse to blindly imitating these imperfect CoTs. The ExPO training objective includes a reinforcement term with an advantage-weighted update, which continues to explore around these initial CoTs. This ensures that the model does not simply memorize the self-generated explanation $\tilde{c}$, but instead uses it as an anchor for exploration, gradually refining the reasoning policy through trial-and-error. On the other hand, we could employ an annealing schedule for the $\beta$ coefficient: gradually decrease $\beta$ as training progresses. This would make the model rely less on the SFT term in later stages, effectively "fine-tuning" its bootstrapped knowledge during warm-up while reducing the risk of locking in any emergent hallucinations.

Looking ahead, we see two critical areas for enabling complex reasoning capabilities through self-improvement: refining algorithmic approaches and optimizing data curation strategies. From an algorithmic perspective, while RL-based training methods are currently seen as a promising direction for achieving strong reasoning abilities, further investigation into their limitations and underlying mechanisms is necessary. For instance, the distribution-sharpening phenomenon observed in GRPO highlights new challenges that need to be addressed for developing truly capable models. From a data perspective, our findings unexpectedly show that expert-generated chain-of-thought examples are not always optimal. This suggests that simply increasing the quality of data does not automatically lead to better performance. Therefore, future efforts should go beyond the focus on expert-annotated high-quality data. Instead, tailoring data to the model's current capabilities and designing learning curricula will be crucial for continued model development.

## Acknolwedgement

This research was supported in part by a research grant from Coefficient Giving. We thank Victor Wang for his valuable discussions and feedback.

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

## A   Details of Property Derivations

Details of injecting the $\nabla_\theta J(\theta)$ and $\boldsymbol{g}(\bar{q}, \bar{c}, \bar{a})$ into the objective3.2 and transforming the expectation into summation. From Equation (3), we have

$$\nabla_\theta J(\theta)^\top \boldsymbol{g}(\bar{q}, \bar{c}, \bar{a}) = \mathbb{E}_{(q, a^\star) \sim \mathcal{D}, (c, a) \sim \pi_\theta(\cdot | q)} \left[ \boldsymbol{g}(\bar{q}, \bar{c}, \bar{a})^\top \nabla_\theta \log \pi_\theta(c, a | q) r(q, c, a, a^\star) \right]$$

$$= \sum_{q, c, a} \mathbb{P}(q) \pi_\theta(c, a | q) \boldsymbol{g}(\bar{q}, \bar{c}, \bar{a})^\top \nabla_\theta \log \pi_\theta(c, a | q) r(q, c, a, a^\star). \qquad (8)$$

From (8), we obtain that

$$\nabla_\theta \boldsymbol{J}(\theta)^\top \boldsymbol{g}(\bar{q}, \bar{c}, \bar{a}) = \underbrace{w \mathbb{P}(\bar{q}) \pi_\theta(\bar{c}, \bar{a} | \bar{q}) \| \nabla_\theta \log \pi_\theta(\bar{c}, \bar{a} | \bar{q}) \|^2 r(\bar{q}, \bar{c}, \bar{a}, a^\star)}_{T_1 \colon \text{ when } (q, c, a) = (\bar{q}, \bar{c}, \bar{a})}$$

$$+ \underbrace{\sum_{q, c, a \neq (\bar{q}, \bar{c}, \bar{a})} w \mathbb{P}(q) \pi_\theta(c, a | q) \nabla_\theta \log \pi_\theta(\bar{c}, \bar{a} | \bar{q})^\top \nabla_\theta \log \pi_\theta(c, a | q) r(q, c, a, a^\star)}_{T_2 \colon \text{ when } (q, c, a) \neq (\bar{q}, \bar{c}, \bar{a})}. \qquad (9)$$

argue that the magnitude of the policy improvement $\Delta \boldsymbol{J}$ (and the alignment between the true gradient $\nabla_\theta \boldsymbol{J}(\theta)$ and the actual used gradient $\boldsymbol{g}(\bar{c}, \bar{a}, \bar{q})$) is dominated by the term $T_1$.

To be more precise, we characterize when $T_2$ (the sum of cross terms $\langle \nabla_\theta \log \pi_\theta(\bar{c}, \bar{a} | \bar{q}), \nabla_\theta \log \pi_\theta(c, a | q) \rangle$ in (9)) is negligible. Intuitively, these gradients lie in extremely high-dimensional spaces (e.g., on the order of billions of dimensions when performing full parameter updates, even on relatively small reasoning models). As a result, unless the questions and corresponding CoTs are highly similar, it is unlikely that the cross-term gradient inner products contribute significantly. In a simplified setting where the logits of the softmax policy $\pi_\theta$ have orthogonal gradients with equal norm, we show that for training samples to which the model assigns lower probability, i.e., $\pi_\theta(\bar{c}, \bar{a} | \bar{q}) < \frac{1}{2}$, the cross-term gradient inner products decrease as the sample becomes less likely. We formalize this below.

**Lemma 2.** *Let $\mathcal{T} = \{(q_j, c_j, a_j)\}_{j=1}^L$ be a finite set, and the policy being a softmax policy over this set, i.e., $\pi_j := \pi_\theta(c_j, a_j | q_j) = \exp(z_j) / \sum_{l=1}^L \exp(z_l)$ where $z_j := f_\theta(q_j, c_j, a_j)$. Assume the following conditions hold: (1) For all $j \neq l$, the gradients of the logits are orthogonal: $\langle \nabla_\theta z_j, \nabla_\theta z_l \rangle = 0$. (2) All logits have the same gradient norm: $\| \nabla_\theta z_j \|^2 = C > 0$ for all $j \in [L]$. Then for any pair $j \neq j'$, the cross-term gradient inner product $\langle \nabla_\theta \log \pi_j, \nabla_\theta \log \pi_{j'} \rangle$ is strictly decreasing in $\pi_j$ if and only if $\pi_j < \frac{1}{2}$.*

The above lemma implies that when the sample $(\bar{q}, \bar{c}, \bar{a})$ is assigned low probability under the current policy (i.e., when $\pi_\theta(\bar{c}, \bar{a} | \bar{q})$ is small), the cross-term contributions in $T_2$ become negligible. Simultaneously, the leading term $T_1$ in (9) is also small in magnitude, as it scales proportionally with $\pi_\theta(\bar{c}, \bar{a} | \bar{q})$. As a result, such samples contribute minimally to overall policy improvement.

In summary, the policy improvement $\Delta \boldsymbol{J}$ will be large on a gradient step with training data $(\bar{q}, \bar{c}, \bar{a})$ when $\pi_\theta(\bar{c}, \bar{a} | \bar{q})$ is high—that is, when the sample is likely under the current policy—and when the corresponding reward is also high.

## B   Proofs

*Proof of Lemma 2.* Let $z_j = f_\theta(q_j, c_j, a_j)$ and define the log-probability gradient as:

$$\nabla_\theta \log \pi_j = \nabla_\theta z_j - \sum_{l=1}^L \pi_l \nabla_\theta z_l =: \nabla_\theta z_j - \bar{z}.$$

Then the inner product becomes:

$$\langle \nabla_\theta \log \pi_j, \nabla_\theta \log \pi_{j'} \rangle = \langle \nabla_\theta z_j - \bar{z}, \nabla_\theta z_{j'} - \bar{z} \rangle.$$

Expanding the above and using the assumptions, we get:

$$\langle \nabla_\theta \log \pi_j, \nabla_\theta \log \pi_{j'} \rangle = \langle \nabla_\theta z_j, \nabla_\theta z_{j'} \rangle - \sum_l \pi_l \langle \nabla_\theta z_j, \nabla_\theta z_l \rangle - \sum_l \pi_l \langle \nabla_\theta z_{j'}, \nabla_\theta z_l \rangle$$

$$+ \sum_{l,m} \pi_l \pi_m \langle \nabla_\theta z_l, \nabla_\theta z_m \rangle$$

$$= 0 - \pi_j C - \pi_{j'} C + \sum_{l=1}^{L} \pi_l^2 C$$

$$= C \left( -\pi_j - \pi_{j'} + \sum_{l=1}^{L} \pi_l^2 \right).$$

To study the monotonicity of the gradient inner product in $\pi_j$, we check the gradient:

$$\frac{d}{d\pi_j} \langle \nabla_\theta \log \pi_j, \nabla_\theta \log \pi_{j'} \rangle = -C + C \cdot \frac{d}{d\pi_j} \left( \sum_l \pi_l^2 \right) = -C + 2C\pi_j = C(2\pi_j - 1).$$

Therefore, the inner product decreases in $\pi_j$ if and only if $\pi_j < \frac{1}{2}$. $\qquad\square$

*Proof of Lemma 1.* We begin by expanding the expectation under the left hand side of the inequality:

$$\mathbb{E}_{\tilde{c} \sim \pi_\theta(\tilde{c}|q,a^*)} \left[ \pi_\theta(a^\star \mid \tilde{c}, q) \right] = \sum_{\tilde{c}} \pi_\theta(\tilde{c} \mid q, a^\star) \cdot \pi_\theta(a^\star \mid \tilde{c}, q)$$

$$= \sum_{\tilde{c}} \frac{\pi_\theta(a^\star \mid \tilde{c}, q) \cdot \pi_\theta(\tilde{c} \mid q)}{\pi_\theta(a^\star \mid q)} \cdot \pi_\theta(a^\star \mid \tilde{c}, q) = \frac{1}{\pi_\theta(a^\star \mid q)} \sum_{\tilde{c}} \pi_\theta(\tilde{c} \mid q) \cdot (\pi_\theta(a^\star \mid \tilde{c}, q))^2$$

$$= \frac{1}{\pi_\theta(a^\star \mid q)} \mathbb{E}_{c \sim \pi_\theta(c|q)} \left[ (\pi_\theta(a^\star \mid c, q))^2 \right],$$

where the second equality holds because of Bayes' rule. Note that after applying the Bayes rule, $\pi_\theta(a^\star \mid \tilde{c}, q)$ cannot be directly generated by the language model, but it is a general distribution (the reverse conditional).

Recall that the right hand side of the inequality is:

$$\mathbb{E}_{c \sim \pi_\theta(c|q)} \left[ \pi_\theta(a^\star \mid c, q) \right] = \sum_c \pi_\theta(c \mid q) \pi_\theta(a^\star \mid c, q) = \pi_\theta(a^\star \mid q).$$

Thus, the inequality we want to prove is reduced to comparing between:

$$\frac{\mathbb{E}[X^2]}{\mathbb{E}[X]} \quad \text{vs.} \quad \mathbb{E}[X] \quad \text{where } X = \pi_\theta(a^\star \mid c, q)$$

By Jensen's inequality (since the square function $x^2$ is convex), we have:

$$\mathbb{E}[X^2] \geq (\mathbb{E}[X])^2 \Rightarrow \frac{\mathbb{E}[X^2]}{\mathbb{E}[X]} \geq \mathbb{E}[X],$$

which completes the proof. $\qquad\square$

## C  More discussion on probability shifts

During RL-type training like GRPO and iterative DPO, negative samples are on-policy and have relatively high probability under the current policy. During training, the probability of these samples

are pushed down, while the probability of the positive samples will be pushed up. We provide a discussion on how pushing up samples with different probability (under the current policy) influences the change of other sample's probabilities when the policy is a softmax of independent logits. The key takeaways are: (1) Pushing up samples that have low probability under the current policy will not make minimal changes to other samples' probabilities (thus the in-distribution property is needed for effective training). (2) Pushing up and down equal amounts have different effects: the magnitude of changes on other logits depend on the magnitude of the original logits.

To be more specific, let $\pi_k = \frac{e^{z_k}}{\sum_m e^{z_m}}$ be a softmax distribution over logits $\{z_k\}_{k=1}^n$, and define a perturbation:

$$z_i' = z_i + \Delta_{\text{up}}, \quad z_j' = z_j - \Delta_{\text{down}}, \quad z_k' = z_k \text{ for } k \neq i, j.$$

Let $\alpha = e^{\Delta_{\text{up}}}$, $\beta = e^{-\Delta_{\text{down}}}$, and define:

$$Z = \sum_m e^{z_m}, \quad Z' = \alpha e^{z_i} + \beta e^{z_j} + \sum_{k \neq i, j} e^{z_k}.$$

Then the change in softmax probabilities $\Delta \pi_k = \pi_k' - \pi_k$ is given by:

$$\Delta \pi_i = e^{z_i} \left( \frac{\alpha}{Z'} - \frac{1}{Z} \right),$$

$$\Delta \pi_j = e^{z_j} \left( \frac{\beta}{Z'} - \frac{1}{Z} \right),$$

$$\Delta \pi_k = e^{z_k} \left( \frac{1}{Z'} - \frac{1}{Z} \right) \quad \text{for } k \neq i, j.$$

First, note that $\alpha > 1$ and $\beta < 1$. Thus, $\Delta \pi_i > 0$ and $\Delta \pi_j < 0$. The change of other logits are relative to their rank: $\Delta \pi_k$ is proportional to $e^{z_k}$.

Thus, we have:

- Increasing $z_k$ (pushing up sample $k$) increases $\pi_k$ and decreases $\pi_i$ for all $i \neq k$.

- Decreasing $z_k$ (pushing down sample $k$) decreases $\pi_k$ and increases $\pi_i$ for all $i \neq k$.

- The magnitude of these changes is proportional to the product of the original probabilities, implying that updates to high-probability samples have larger impact than those to low-probability candidates.

- The effect of only pushing up/down sample probabilities differs from simultaneously pushing up positive samples and pushing down negative ones. Relying solely on negative gradients (i.e., providing signals to decrease the probability of certain samples) will not be effective unless the model already assigns high probability to the correct outputs. This approach is especially ineffective for problems where the model is only partially correct. Conversely, only pushing probabilities up of certain samples can lead to overly aggressive updates. Therefore, it's important to include both appropriate positive and negative samples.

## D  Additional Result

### D.1  Detailed Experimental settings

In the ExP-DPO experiments, we train LLaMA-3.2-3B-instruct and QWEN-2.5-3B-instruct on a single NVIDIA H100 GPU. The optimizer is AdamW with cosine learning rate scheduler and 0.05 warmup ratio, where the maximum learning rate is 5e-7. The training batch size is 16. The basic code frameworks are the trl library [30] `https://github.com/huggingface/trl` and openr1 [4] `https://github.com/huggingface/open-r1`. The evaluation is performed under the zero-shot prompting, with the simplest prompt for "think step by step" and the answer format. We run the evaluation for 4 passes and calculate the match via MathVerify [17] `https://github.com/huggingface/Math-Verify`.

Similarly, in the ExP-GRPO experiments, we fine-tune LLaMA-3.2-3B-instruct and QWEN-2.5-3B-instruct based on the X-R1 [10] `https://github.com/dhcode-cpp/X-R1` GRPO trainer on 4

NVIDIA A100 GPUs (80GB each). Training is performed for 3 epochs with a per-device batch size of 3 and a gradient accumulation step size of 8. We adopt the AdamW optimizer with a cosine learning rate schedule, setting the maximum learning rate to 3e-6 and a warmup ratio of 0.1. Mixed-precision training is enabled via bfloat16, and flash attention v2 is used to accelerate attention computation. Gradient checkpointing is applied to reduce memory usage, and training is conducted using the accelerate framework with ZeRO Stage 3 configuration. For both experiments, during training, the generation temperature is set to 0.9, while for evaluation it is set to 0.7.

### D.2 Outputs at Initialization

The expert-annotated CoT is mathematically dense and concise, demonstrating the solution by expanding and cancelling terms in a single step. In contrast, the self-explanation is more pedagogical and compositional; it first defines the core concept, then explicitly states the key logical insight ($9! = 9 \times 8!$), and finally performs the simplified calculation. This decomposed, principle-first approach provides a better in-distribution learning signal for RL training.

---

**Example of expert-annotated CoT vs explanation for the question from MATH**

**Question**:
Compute without using a calculator: $9!/8!$

**Expert-annotated CoT**:
$9!/8! = \frac{9 \times 8 \times \cdots \times 1}{8 \times 7 \times \cdots \times 1} = \boxed{9}$

**Self-explanation**:
The expression $9!/8!$ can be simplified by recognizing that the factorial function $n!$ (n factorial) is defined as the product of all positive integers up to $n$. So $9! = 9 \times 8 \times \cdots \times 1$ and $8! = 8 \times 7 \times \cdots \times 1$. Therefore, $9! = 9 \times 8!$. When we divide $9!$ by $8!$, we get:

$$\frac{9!}{8!} = \frac{9 \times 8!}{8!} = 9$$

Hence, $9!/8! = \boxed{9}$.

---

We also provide the full example of direct CoT vs explanation for the question from MATH in Subsection 4.1.

---

**Full example of direct CoT vs explanation for the question from MATH**

**Question**:
A box contains six cards. Three of the cards are black on both sides, one card is black on one side and red on the other, and two of the cards are red on both sides. You pick a card uniformly at random from the box and look at a random side. Given that the side you see is red, what is the probability that the other side is red? Express your answer as a common fraction.

**Direct CoT**:
To solve this problem, we'll break it down into steps.
Step 1: Count the total number of sides that are red. - There are 2 cards that are red on both sides, so each has 2 red sides. That's 2 * 2 = 4 red sides. - There is 1 card that is black on one side and red on the other, so it has 1 red side. - Total red sides = 4 + 1 = 5
Step 2: Determine the total number of sides in the box. - There are 3 cards that are black on both sides, so each has 2 sides. That's 3 * 2 = 6 sides. - There are 2 cards that are red on both sides, so each has 2 sides. That's 2 * 2 = 4 sides. - Total sides = 6 + 4 = 10
Step 3: Calculate the probability that the other side is red given that the side you see is red. - There are 5 red sides in total, and 5 of those are from cards that are red on both sides. - Since we see a red side, we have 5 options for the side we see, and 5 of those are from cards that are red on both sides. - The probability that the other side is red given that the side you see

---

is red is the number of red sides from cards that are red on both sides divided by the total number of red sides. - Probability = Number of red sides from cards that are red on both sides / Total red sides = 5 / 5 = 1

Therefore, the probability that the other side is red given that the side you see is red is $\boxed{1}$.

**Explanation**:

To solve this problem, we first need to find the total number of red sides on all the cards in the box.

There are two cards that are red on both sides, so they have 4 red sides in total. There is one card that is black on one side and red on the other, so it has 1 red side.

In total, there are 4 + 1 = 5 red sides.

Now, we need to find the probability that the other side is red given that the side we see is red. This is a conditional probability problem.

There are 5 red sides out of a total of 6 sides in the box.

So, the probability that we see a red side is 5/6.

If we see a red side, it could be from the 4 red sides of the cards that are red on both sides, or from the 1 red side of the card that is black on one side and red on the other.

The 4 red sides of the cards that are red on both sides are out of the 5 red sides we see.

So, the probability that the other side is red, given that we see a red side, is 4/5.

The answer is based on the idea of conditional probability. We know that there are 5 red sides out of a total of 6 sides in the box. When we see a red side, it could be from 4 different cards, each with 2 red sides, or from 1 card with 1 red side. Therefore, the probability that the other side is red, given that we see a red side, is 4 out of the 5 red sides.

## D.3 Results for ExP-DPO

|  | Offline LLaMA3.2-3B-Instruct | | Online LLaMA3.2-3B-Instruct | | Offline Qwen2.5-3B-Instruct | | Online Qwen2.5-3B-Instruct | |
|---|---|---|---|---|---|---|---|---|
| Version | $\tilde{c}_+$ | $c_{E_+}$ | $\tilde{c}_+$ | $c_{E_+}$ | $\tilde{c}_+$ | $c_{E_+}$ | $\tilde{c}_+$ | $c_{E_+}$ |
| MATH | 45.7 | 38.7 | 50.2 | 53.6 | 54.3 | 43.7 | 60.4 | 49.3 |
| GSM8K | 71.2 | 63.7 | 81.5 | 74.4 | 80.1 | 69.6 | 85.4 | 76.3 |

Table 3: Pass@4 performance on MATH and GSM8K with models trained with ExP-DPO $\tilde{c}_+$ and expert CoT $c_{E_+}$. ExP-DPO consistently outperformed $c_E$ trained policy in both online and offline settings.

We evaluate two variants of ExP-DPO: an offline and an iteratively updated online version. For offline, we first compare the impact of using expert CoTs ($c_E$) versus self-generated explanations ($\tilde{c}$) as the preferred response. As shown in Table 3, training with $c_E$ leads to only marginal improvements in downstream performance. This observation aligns with our earlier analysis: although expert CoTs are semantically correct, they often lie in low-probability regions under the current policy $\pi_\theta$, violating the in-distribution criterion (Property 1). Consequently, they induce misaligned gradient signals. Moreover, from visualization of its loss (see Appendix for details), we observe that even minimal increases in the log-probability of $c_E$ are sufficient to yield a near-zero loss, creating the illusion of rapid convergence. Therefore, we transition to the online ExP-DPO setting, where the preferred explanation $\tilde{c}$ is regenerated periodically from the current policy. These improvements underscore the importance of maintaining distributional alignment between positive training samples and the evolving policy. By continuously adapting $\tilde{c}$ to reflect the model's current beliefs, ExP-DPO ensures that learning remains anchored in the regions where the model can most effectively improve, thereby enabling robust optimization without relying on external expert demonstrations.

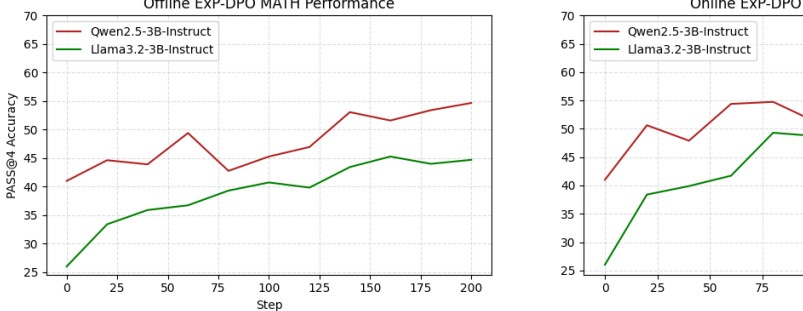

Figure 4: We compare ExP-DPO performance on the MATH dataset across training steps for two base models: Qwen2.5-3B-Instruct and Llama3.2-3B-Instruct. Online ExP-DPO consistently outperforms its offline counterpart, confirming that updating the explanation-based positive samples improves learning efficiency and final accuracy. Qwen2.5 shows higher sample efficiency and peak accuracy than Llama3.2 under both settings.

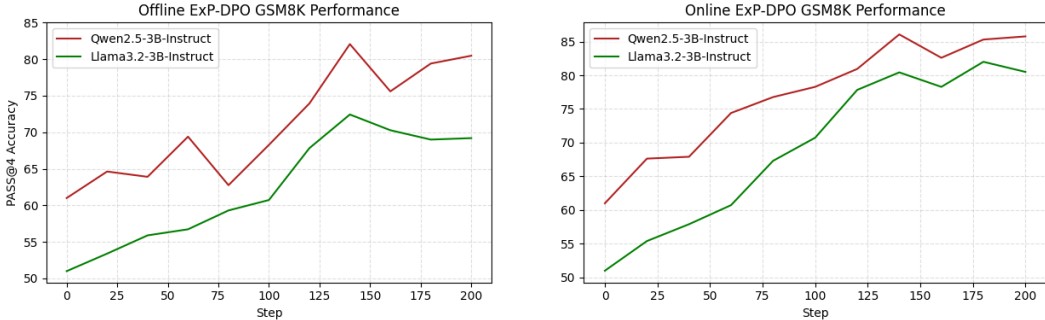

Figure 5: ExP-DPO performance on the GSM8K dataset for Qwen2.5-3B-Instruct and Llama3.2-3B-Instruct models. Online ExP-DPO achieves stronger performance and faster convergence compared to the offline setting. Qwen2.5 benefits more from the online explanation updates, attaining over 85% accuracy, while Llama3.2 saturates earlier.

## D.4 Deceptive Loss Dynamics in DPO Training

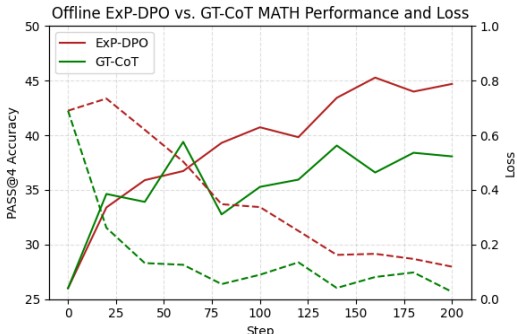

Figure 6: Training curves on the MATH dataset using LLaMA-3.2-3B-Instruct. We compare standard DPO (with ground-truth answer as the preferred response) against our ExP-DPO method. The solid lines denote model performance (PASS@4 accuracy), while the dashed lines represent the loss. We observe that under standard DPO, the training loss rapidly decreases to a low value; however, this does not translate into improved downstream performance, revealing a phenomenon of *deceptively low loss*. In contrast, ExP-DPO exhibits a smoother loss decay and leads to significantly higher task accuracy. This demonstrates that ExP-DPO provides a more reliable learning signal and avoids the overfitting or shortcut behaviors often associated with poorly aligned positive preference supervision.

## D.5 Choice of $\beta$ for ExP-GRPO

In our implementation of ExP-GRPO, the guidance term of self-explanations on the ground-truth answer is incorporated into the overall objective via a scaling coefficient $beta$: $\beta \log \pi_\theta(\tilde{c}, a^\star | q)$ This term is designed to provide an additional learning signal when the model's own sampled responses fail to yield effective gradients, particularly in challenging reasoning settings. We fix $\beta = 0.04$ in all reported experiments. To justify this design, we conducted an ablation study across multiple $\beta$ values on the full MATH training set (1 epoch).

| Model \ $\beta$ | 0.5 | 0.1 | 0.08 | 0.04 | 0.01 | 0 |
|---|---|---|---|---|---|---|
| Qwen2.5-3B-Instruct | 47.3% | 48.6% | 50.5% | 68.7% | 60.4% | 51.6% |
| LLaMA-3.2-3B-Instruct | 33.5% | 34.8% | 35.3% | 58.9% | 46.6% | 44.3% |

Table 4: Accuracy of ExPO-GRPO with different $\beta$ values on the MATH training set after 1 epoch of training. We can see that $\beta = 0.04$ yields to the best training result.

From Table 4, we observe a clear trend: the addition of the ExP-SFT term significantly boosts performance over the baseline ($\beta = 0$), particularly in early-stage training where standard GRPO fails to provide meaningful updates. However, excessively large $\beta$ values (e.g., $\beta = 0.5$) degrade performance, likely due to over-reliance on imperfect guidance signals. In contrast, smaller $\beta$ values (e.g., $\beta = 0.01$) yield slower learning.

Thus, $\beta = 0.04$ offers a favorable trade-off between initial learning efficiency and final performance. It balances the model's ability to leverage the structured supervision from self-generated explanations without overwhelming the exploration driven by the advantage term in GRPO. These results empirically support our design choice and highlight the complementary role of the ExP-SFT term in bootstrapping reasoning capability where standard policy optimization struggles.

### D.6 ExPO-GRPO results across different base model sizes

|  | Qwen2.5-1.5B-It | LLaMA-3.2-1B-It | Qwen2.5-3B-It | LLaMA-3.2-3B-It | Qwen2.5-7B-It | LLaMA-3.1-8B-It |
|---|---|---|---|---|---|---|
| ExP-GRPO | 45.2% | 40.5% | 68.7% | 58.9% | 83.2% | 70.9% |
| GRPO SFT-GT-CoT | 38.8% | 35.8% | 61.9% | 48.2% | 79.8% | 64.9% |
| GRPO | 32.5% | 28.3% | 50.3% | 44.4% | 78.1% | 63.9% |

Table 5: Accuracy of ExPO-GRPO across different base model sizes on the MATH test set. For >7B model, we use the LoRA parameter-efficient fine-tuning. ExPO consistently outperforms other baselines across all tested scales.

To better demonstrate the generality and robustness of ExPO, we conducted a broader evaluation across model scales. The results show that ExPO performance gains hold even in >=7B models, suggesting that ExPO's design, which is grounded in in-distribution gradient alignment and positive learning signal, scales effectively with model capacity. These results strengthen our claim that ExPO is a scalable and general reinforcement learning method for reasoning, even in the absence of expert CoT supervision.

### D.7 Additional Result for ExP-GRPO

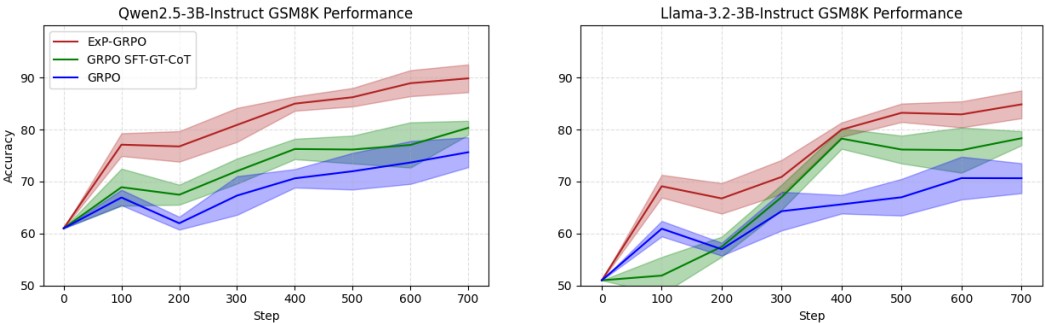

Figure 7: Test accuracy on GSM8K for Qwen2.5-3B-Instruct (left) and LLaMA-3.2-3B-Instruct (right) across global training steps. **ExP-GRPO** consistently surpasses both **GRPO** and **GRPO SFT-GT-CoT**, the latter of which incorporates supervised fine-tuning on expert CoT $c_E$. These results highlight the effectiveness of **ExP-GRPO** in delivering stronger and more generalizable learning signals, resulting in improved sample efficiency and superior final performance.

