# OpenReview forum: "ExPO: Unlocking Hard Reasoning with Self-Explanation-Guided Reinforcement Learning"
_NeurIPS.cc/2025/Conference — NeurIPS 2025 poster_

### Official Review · Reviewer_D1iU · 2025-06-06

**Clarity:** 2
**Significance:** 3
**Originality:** 3
**Rating:** 3
**Confidence:** 3

**Summary:**

This paper introduces ExPO (Self-Explanation Policy Optimization), a modular RL method based on DPO or GRPO that can improve LLM learning efficiency and final performance on math reasoning benchmarks. By bootstrapping the LLM with the question and ground truth answer, ExPO obtains positive samples that satisfy two criteria: (1) in-distribution and (2) positive learning signal, which are considered essential for the main optimization objective. ExPO leverages these positive samples to provide more effective learning signals for DPO and GRPO. Through experiments that compare ExPO with expert CoT baselines, this paper demonstrates the effectiveness of ExPO.

**Questions:**

1. Bootstrapping the LLM with the ground truth answer is effective for some questions since it reshapes the search space, allowing the LLM to "think outside the box". However, it may also introduce some bias. For questions that the LLM can not figure out a real correct solution even with the hint (which can be common for complex reasoning questions), it still generates a CoT solution although it's not correct (falsely copying the ground truth answer). Learning from these CoT data may have negative influences. Have you noticed or considered this problem?
2. In the reconstructed GRPO objective, the scale of the original part and the additional SFT term seem unclear, making the new objective design seem unreliable. Are there any justifications or observations that support this design?
3. As mentioned in the Weaknesses part, could you elaborate on the method you use to obtain expert CoT data, as well as the compared GRPO SFT-GT-CoT baseline?
4. Are there any RL results for models at different scales (7B, 32B, etc.)? Considering that currently displayed results are all for 3B LLMs, it may not reflect the complete picture.

**Ethical Concerns:**

["NO or VERY MINOR ethics concerns only"]

**Final Justification:**

I have raised my score for the author's supplementary experimental results, which improved the solidity of this paper. However, the proposed method may introduce bias or cause hallucination in a long term perspective, since it relies on self-generated explanations. Besides, the generalizability of this method towards other domains needs further experimentation.

**Limitations:**

1. As mentioned in the Weaknesses, the generalizability of this approach is limited.

**Quality:**

1

**Strengths And Weaknesses:**

- **Strengths**
1. Significance: This paper provides some insights in the selection of training data and the optimization process of reinforcement learning algorithms, discovering a new approach for improving current RL algorithms like GRPO and DPO.
2. Originality: The idea of incorporating STaR-style rationales [1] into RL optimization is novel. This paper seems to have made an effective attempt on extending the STaR methodology.
- **Weaknesses**
1. Rigor: One major issue of this paper is insufficient rigor concerning mathematical derivations and conclusions. For example, the claim in expression (7) is not supported by convincing mathematical proofs (Lemma 1 is actually very weak). The conclusion that data with higher probability leads to greater gradient is also erroneous empirically.
2. Insufficient experimentation: The displayed experimental results are very limited. It can not suffice to justify the ExPO approach.
3. Clarity: This paper also faces clarity issues. For example, the source of expert CoT data is unclear, although it is considered as an important baseline. Besides, The implementation of the GRPO SFT-GT-CoT baseline is also ambiguous.
4. Generalizability: The proposed bootstrapping method based on ground truth answer may not be applicable for other reasoning tasks such as code generation.

[1] Zelikman, E., Wu, Y., Mu, J., & Goodman, N. (2022). Star: Bootstrapping reasoning with reasoning. Advances in Neural Information Processing Systems, 35, 15476-15488.

---

> ### Author Rebuttal · Authors · 2025-07-31
>
> We thank the reviewer for their thoughtful comments and appreciate the recognition of our contributions to training data selection and RL optimization, as well as the novelty of incorporating STaR-style rationales into RL.
>
> The core contribution of our paper is **ExPO**, a method to bootstrap learning in **hard reasoning tasks** where positive samples are rare (e.g., Qwen2.5-3B achieves only 4% pass@64 on MATH Level-5). In such settings, GRPO often fails (Fig. 3, Table 2). ExPO addresses this by using **self-explanations** on the correct answer to warm-start training. Empirically, these signals are more effective than expert chain-of-thoughts (CoTs) for hard problems.
>
> To understand why, we build on the open question in STaR about when and why self-explanations[rationales] help. Our **empirical analysis** (Sec. 3.2, App. C.1) shows that: (1) self-explanations have higher likelihood under the current policy than expert CoTs (Fig. 1), and (2) they contain more correct reasoning steps than self-generated CoTs (Fig. 2). Our theory (Sec. 3.1) provides **intuition** on why these properties support effective policy improvement.
>
> We now address each reviewer's point in detail below:
>
> >**W1**: “One major… empirically.”
>
> Thank you for the question. Sec. 3.1 is intended to provide intuition for why the two key properties of self-explanations (identified in Sec. 3.2) may lead to better policy improvement, compared to self-generated and expert CoT. L139-134 (incl. Eq. 7) and Lemma 1 aim to provide intuition on why when a sample has low likelihood, its gradient may have low inner product with the true gradient. We do not claim that high-likelihood samples have high gradient norms, and we will clarify this in the revision.
>
> >**W2 ^ Q4**: “Insufficient …approach.” “Are there … complete picture.”
>
> We thank the reviewer for raising this concern. We agree that a broader evaluation across model scales will better demonstrate the generality and robustness of ExPO. While our original submission focused on 3B-scale models due to resource constraints in academia settings (maximum available compute: 4×L40S, 48GB each), we took the reviewer’s comment seriously and conducted additional experiments under these constraints.
>
> To scale to ≥7B models, we adopted parameter-efficient fine-tuning using LoRA (rank 32) with DeepSpeed ZeRO-3. Despite hardware limitations, this allowed us to run controlled RL training experiments on larger models such as LLaMA-3.1-8B-Instruct and Qwen2.5-7B-Instruct.
>
>
> | Method\Model   | Qwen2.5-1.5B-Instruct | LLaMA-3.2-1B-Instruct | Qwen2.5-3B-Instruct | LLaMA-3.2-3B-Instruct | Qwen2.5-7B-Instruct (LoRA)| LLaMA-3.1-8B-Instruct (LoRA)|
> |-|-|-|-|-|-|-|
> | ExP-GRPO | 45.2%     | 40.5%     | 68.7%     | 58.9%     | 83.2%     | 70.9%     |
> | GRPO SFT-GT-CoT | 38.8%     | 35.8%     | 61.9%     | 48.2%     | 79.8%     | 64.9%     |
> | GRPO | 32.5%     | 28.3%     | 50.3%     | 44.4%     | 78.1%     | 63.9%     |
>
> ExPO consistently outperforms other baselines on MATH[2] across all tested scales, particularly on difficult questions (e.g., MATH[2] level-5 from table 2 of our paper). Notably, the performance gains hold even in >=7B models, suggesting that ExPO's design—grounded in in-distribution gradient alignment and positive learning signal—scales effectively with model capacity. These results strengthen our claim that ExPO is a scalable and general reinforcement learning method for reasoning, even in the absence of expert CoT supervision.
>
> We believe this addresses the reviewer’s concern regarding the scope of the experimental validation. Furthermore, we are committed to expanding these large-scale experiments and will include more results in the future version of the paper, where possible.
>
> >**W3a & Q3**: “The source … GRPO SFT-GT-CoT baseline?”
>
> Thank you for raising this point.. The expert CoT annotations used in our baselines for GSM8K and MATH are sourced from the official dataset releases associated with the following references:
>
> *[1]Cobbeet al. Training verifiers to solve math word problems. 2021.
> *[2]Hendrycks et al. Measuring mathematical problem solving with the math dataset, 2021.
>
> These datasets are publicly available and widely adopted in prior work on mathematical reasoning. Due to the rebuttal policy, we cannot include direct hyperlinks, but we confirm that these datasets (including expert CoTs) are accessible through their official HuggingFace repositories. We have also ensured that our implementation uses these expert CoTs exactly as provided in the original dataset releases, without any modifications.
>
> We will clarify this in the revision to ensure transparency and reproducibility.
>
> >**W3b**: “The implementation … ambiguous.“
>
> We apologize for the ambiguity and thank the reviewer for pointing this out. To clarify, the GRPO SFT-GT-CoT baseline corresponds to a modified version of the GRPO objective described in Sec. 3.3.2 (ExP-GRPO). Specifically, in the ExP-GRPO formulation, the ExP-SFT term:
>
> $
> \beta \log \pi_\theta(\tilde{c},a^{\star} | q)
> $
>
> (where $\tilde{c}$ is the self-explanation on the ground-truth answer) is replaced in the GRPO SFT-GT-CoT baseline by:
>
> $
> \beta \log \pi_\theta(c_{E},a^{\star} | q)
> $
>
> where $c_{E}$ is the expert CoT annotation provided in the official release of the MATH or GSM8K datasets. All other components of the GRPO training objective remain the same. We will make this detail more explicit in the revision to prevent confusion.
>
> >**W4**: “Generalizability … generation.“
>
> Thank you for raising this point. While our experiments focus on math reasoning, the core idea of ExPO—bootstrapping learning with self-generated, in-distribution samples conditioned on verifiable outcomes—applies broadly. For example, in code generation (e.g., Codeforces), correct outputs can be verified via test cases. Given the desired output, the model can be prompted to generate rationale (e.g., pseudocode), enabling ExPO to provide effective training signals, as discussed in Sec. 3. More generally, any reasoning task with verifiable rewards—such as physics or scientific QA—can benefit from this approach. We agree that applying ExPO in these domains may require tailored prompting or tools, and we see this as a promising direction for future work. We will clarify this in the final version.
>
> >**Q1**: “Generates … problem?”
>
> We thank the reviewer for this insightful question. Indeed, self-explanations conditioned on the ground-truth answers (denoted $\tilde{c} \sim \pi_\theta ( \cdot | q, a^\star)$) are not guaranteed to be logically correct or aligned with optimal reasoning. We clarify both their intent and role in the learning process.
>
> **The core goal of our method is not to teach perfect reasoning, but to jump-start learning when standard RL objectives (e.g., GRPO) fail to provide gradients—especially on hard questions where models can't produce correct responses.** In such cases, $\tilde{c}$ often provides partial or heuristic reasoning that:
>
> * Increases the likelihood of predicting the correct answer (Property 2: Positive Learning Signal), and
> * Is more in-distribution than expert CoTs under the current policy (Property 1).
>
> Importantly, ExPO does not blindly imitate these imperfect CoTs. The advantage-weighted update continues exploration around them, allowing the model to refine its reasoning policy through trial-and-error.
>
> **As shown in Sec. 4.2 and Table 2, this mechanism is highly effective on MATH level-5 questions where other methods fail.** ExPO enables bootstrapped learning and continued improvement, showing that imperfect CoTs can still serve as valuable scaffolds.
>
> We agree this is an important point and will clarify the exploratory role of the advantage-weighted term in the final version.
>
> >**Q2**: “In … design?”
>
> We thank the reviewer for the question and apologize for the lack of clarity in our description. In our implementation of ExP-GRPO, the guidance term—derived from self-explanations on the ground-truth answer—is incorporated into the overall objective via a scaling coefficient $\beta$:
>
> $
> \beta \log \pi_\theta(\tilde{c},a^{\star} | q)
> $
>
> This term is designed to provide an additional learning signal when the model’s own sampled responses fail to yield effective gradients, particularly in challenging reasoning settings. We fix $\beta = 0.04$ in all reported experiments. To justify this design, we conducted an ablation study across multiple $\beta$ values on the full MATH[2] training set (1 epoch):
>
> | $\beta$ \ Model | Qwen2.5-3B-Instruct | LLaMA-3.2-3B-Instruct |
> |-|-|-|
> | 0.5 | 47.3%     | 33.5%     |
> | 0.1 | 48.6%     | 34.8%     |
> | 0.08 | 50.5%     | 35.3%     |
> | **0.04** | **68.7%**     | **58.9%**     |
> | 0.01 | 60.4%     | 46.6%     |
> | 0.0 | 51.6%     | 44.3%     |
>
> (Pass@4 accuracy is reported on MATH[2] test set)
>
> We observe a clear trend: the addition of the ExP-SFT term significantly boosts performance over the baseline ($\beta = 0$), particularly in early-stage training where standard GRPO fails to provide meaningful updates. However, excessively large $\beta$ values (e.g., $\beta = 0.5$) degrade performance, likely due to over-reliance on imperfect guidance signals. In contrast, smaller $\beta$ values (e.g., $\beta = 0.01$) yield slower learning.
>
> Thus, $\beta = 0.04$ offers a favorable trade-off between initial learning efficiency and final performance. It balances the model’s ability to leverage the structured supervision from self-generated explanations without overwhelming the exploration driven by the advantage term in GRPO. These results empirically support our design choice and highlight the complementary role of the ExP-SFT term in bootstrapping reasoning capability where standard policy optimization struggles.

---

> > ### Comment · Reviewer_D1iU · 2025-08-03
> >
> > Thank you for the detailed response!
> >
> > The authors provided supplementary experimental results concerning different model scales, which addressed my concerns on model generalizability. The ablation study on the scale coefficient $\beta$ demonstrates the validity of the ExP-SFT term, also balancing the original objective part. Therefore, I have raised my score for the author's contribution. However, I still believe that the proposed method may introduce bias or cause hallucination in a long term perspective, since it relies on self-generated explanations. Besides, the generalizability of this method towards other domains needs further experimentation.

---

### Official Review · Reviewer_uMrA · 2025-06-30

**Clarity:** 3
**Significance:** 3
**Originality:** 3
**Rating:** 4
**Confidence:** 4

**Summary:**

This paper focuses on RL-style post training method, and deals with high cost and sparsity of positive responses. The authors identify two key properties of effective positive samples, high probability under the current policy and increase the model's likelihood for the correct answer. And then, this paper develops the self-explanation policy optimization, ExPO, which generates such positive samples from the model itself. Experiments confirm the effectiveness and efficiency.

**Questions:**

1.The paper argues in introduction that "Perhaps surprisingly, empirically, we observe that these "gold" responses—though containing correct CoTs—may fail to provide effective learning signals". I suggest adding such empirical experiments in the intro to make the motivation clearer.

2.Like DeepSeek-R1, only relying the system prompt (<think>) and ground-truth answer can start RL training. Does it conflict with the claims (e.g., in Line 50 "Unlike prior work that relies on fixed expert CoTs") in this paper?

3.Some details can be enhanced, 1) in figure 1-2, the base model is qwen2, while the main experiments use qwen2.5, what is the reason behind it? 2) in table 1, can you provide a detailed introduction on the offline and online?

4.Typos: 1) Line 48: "positive training samples These samples" ->"positive training samples. These samples" 2) Line 227: "self-generated CoT c" -> "self-generated CoT c˜"

**Ethical Concerns:**

["NO or VERY MINOR ethics concerns only"]

**Final Justification:**

Comprehensively considering the original paper's contribution and the rebuttal, I think the original score is appropriate and I tend to keep my score.

**Limitations:**

yes

**Paper Formatting Concerns:**

No major formatting issues

**Quality:**

3

**Strengths And Weaknesses:**

- Strength

1. Continuous improvement and exploration of effective principles for RL on LLM reasoning are of great research significance.

2. This paper provides a comprehensive theoretical analysis for exploring the effectiveness of RL-style post training.

3. The authors analyze both the offline DPO and onlie GRPO to confirm the reliability of ExPO.

- Weakness

1. From the technical perspective, the contribution of this paper is somewhat weak. Generating a self-explanation conditioned on the correct answer in Eq.(8) is not new for existing work.

2. The sampling efficiency is not well consolidated and validated.

3. Some details are not clear, referring to questions.

---

> ### Author Rebuttal · Authors · 2025-07-31
>
> We sincerely thank the reviewer for their time and insightful feedback. We are especially grateful and very encouraged for their recognition on the work's "great research significance," our "comprehensive theoretical analysis," and the experimental reliability confirmed through "both the offline DPO and online GRPO."
>
> > **Weakness 1**: Generating a self-explanation conditioned on the correct answer in Eq.(8) is not new for existing work.
>
> Thank you for this comment. We agree that the concept of generating explanations conditioned on the correct answer and using it for supervised fine-tuning (SFT) has been explored in STaR. However, our work builds directly on that foundation and addresses a deeper, unresolved question that STaR itself identifies.
>
> While STaR demonstrated the empirical effectiveness of self-explanation during SFT, the authors explicitly state that the underlying reasons for this effectiveness remain unclear. To quote from their paper: “Future work should establish more connections between rationalization [self-explanations] and these RL objectives, and examine more generally when and why rationalization [self-explanations] improves learning.” Our work is a direct response to this open problem.
>
> Our primary contribution is not only the act of conditional self-explanation generation, but rather the easy-to-use ExPO framework that integrates these self-explanations into RL training, together with a formalization on what may enable self-explanations to provide better learning signals to language models. We identify two key properties that high-quality positive samples must satisfy. This formalization makes our approach more general and extensible than previous methods: it clarifies that many other self-explanations (e.g., explanations based on additional hints) can serve as a positive sample, as long as they conform to the two fundamental properties we propose.
>
> > **Weakness 2**: The sampling efficiency is not well consolidated and validated.
>
> Thank you for this question regarding the sampling efficiency. In standard GRPO training, only a correctly generated CoT could provide the positive training signal (if we disgard the formatting rewards or if all samples have the same formatting rewards). This process is highly inefficient for difficult tasks. For instance, on MATH Level 5 questions, the qwen-2.5-3b- instruct model achieves only 4% accuracy for pass@64 and only 9% for pass@128. This means over 90% of the computational effort in sampling the model responses for these hard tasks during training may beis wasted  as the responses cannot be used for providing learning signal.
>
> In contrast, as shown in Figure 2, our self-explanation method generates significantly higher-quality CoTs. By producing more valid reasoning paths, a much larger fraction of the sampled responses can be utilized to provide positive signals, making the training process more sample-efficient and computationally effective as shown in the test performance trajectory during training (Figure 3 and 4).
>
> > **Question 1**: The paper argues in introduction that "Perhaps surprisingly, empirically, we observe that these "gold" responses—though containing correct CoTs—may fail to provide effective learning signals". I suggest adding such empirical experiments in the intro to make the motivation clearer.
>
> Thank you for this suggestion. The statement you highlighted is a reference to one of our experimental findings: in the case of math reasoning tasks of MATH and GSM8K, the self-explanation approach can outperform training on "gold" expert-annotated CoTs.
>
> This was at first an unexpected observation during our early experiments, which motivated us to investigate the underlying reasons more deeply. This investigation led to the formulation of the in-distribution property. It emphasizes the importance of tailoring training data to the model's current capabilities rather than simply using human-annotated 100% correct data that may be too far beyond the model's current policy.
>
> Thanks for your comment and we will revise the introduction to make this connection explicit and point the reader to the specific results in the experiment section. We appreciate you helping us make the paper's motivation clearer.
>
> > **Question 2**: Like DeepSeek-R1, only relying the system prompt ([object Object]) and ground-truth answer can start RL training. Does it conflict with the claims (e.g., in Line 50 "Unlike prior work that relies on fixed expert CoTs") in this paper?
>
> Thank you for this question and we want to clarify the context of our work and its primary contribution. Our claim in Line 50, "Unlike prior work that relies on fixed expert CoTs," refers to a common practice in the field. When high-quality, human-annotated CoTs are available, researchers often use them to train reasoning models. This is prevalent in both major training stages:
>
> - In the SFT stage: for example LLaVA-CoT and ReFT use expert-annotated CoTs to train the model; DeepSeek-R1 uses a small dataset of gold CoTs to warm up the model.
>
> - In the RL stage: imitation learning and behavioral cloning methods often depend on expert demonstrations, for example in the BREAD and FingER papers.
>
> Our work addresses the scenario where such expert data is NOT available (or only a few), so that the GRPO method is commonly used, which is also the case for DeepSeek-R1. Our ExP-GRPO experiment is designed to operate in this setting.
>
> Under the "no-expert-data" scenario, our work emphasizes the importance of model self-generated data and formalizing the principles to explain why self-generated data could provide a better signal than expert-annotated data. We formalize this intuition in Property 1 and Figure 1 empirically supports the intuition that self-explanations are indeed more in-distribution than expert CoTs. Furthermore, the experiments of ExP-DPO $c_+$ vs $C_E$ in Table 1 and ExP-GRPO vs GRPO SFT-GT-CoT in Figure 4 show that these in-distribution samples provide effective learning signals and lead to better performance.
>
> > **Question 3**: Some details can be enhanced, 1) in figure 1-2, the base model is qwen2, while the main experiments use qwen2.5, what is the reason behind it? 2) in table 1, can you provide a detailed introduction on the offline and online?
>
> 1)Thank you for your careful reading and this was a typo in the figure captions. The Initial model used throughout all of our experiments is Qwen2.5. We will correct this typo in the revised manuscript.
>
> 2)We explain in detail the offline and online DPO methods here.
>
> - Offline DPO is the classical DPO method. The entire dataset of preference pairs is generated once at the beginning of training using the initial reference policy π_ref. For offline ExP-DPO, both the preferred winning and dispreferred losing responses are sampled from this initial policy, i.e. $\tilde{c_+} \sim \pi_{ref} ( \cdot | q, a^\star)$ and $\tilde{c_-} \sim \pi_{ref} ( \cdot | q )$.
>
> - Iteratively online DPO method has a dynamic training process. At the beginning of each iteration, the preferred winning explanation is regenerated using the current updated policy, as well as the dispreferred samples. This allows the model to learn from progressively better positive examples as it improves.
>
> > **Question 4**: Typos.
>
> We sincerely thank the reviewer for their meticulous attention to detail in identifying several typos. We will correct them in the revised version. We appreciate their effort in helping us improve the overall readability and quality of the paper.

---

> > ### Comment · Reviewer_uMrA · 2025-08-06
> > **Response to Authors**
> >
> > Thanks for your detailed responses which have addressed most my concerns. Comprehensively considering the original paper and the rebuttal, I think the original score is appropriate and I tend to keep my score (Borderline accept).

---

### Official Review · Reviewer_UZkv · 2025-07-02

**Clarity:** 2
**Significance:** 4
**Originality:** 4
**Rating:** 5
**Confidence:** 4

**Summary:**

This paper investigates the properties of chain of thought (CoT) that make them useful in improving the reasoning abilities of models. The main theoretical hypotheses are: 1) CoTs that are in-distribution are more useful than other CoTs, and 2) CoTs that increase the likelihood of the answer are more useful than other CoTs. Formal proofs for these statements are presented. Based on these hypotheses, the authors generate CoTs conditioned on the gold answer, also referred to as "self-explanations," thus meeting both criteria — being in-distribution and increasing the likelihood of producing the target answer. The proposed method ExPO demonstrates that these self-explanations outperform expert CoTs and other CoTs in both offline and online policy optimization using two training methods, DPO and GRPO.

**Questions:**

1. Please explain the statement: "They must be better than negative samples in achieving the task at hand." What does it mean for a negative example to achieve a task?
2. Please clarify: "In settings where a+ and a_ are incorrect"—what's an example of this? Why would anyone train on CoTs that have generated wrong answers in the first place? Neither DPO nor GRPO does this.
3. What happens if the likelihood of the correct answer is high but occurs under a "wrong CoT"? One could easily generate such data to meet both criteria (using an LLM as a judge and selecting incorrect CoTs), and the trained model would likely be worse. Doesn't this contradict your claims even though the conditions are met?
4. What if the self-explanation CoT yields a higher likelihood for a wrong answer, i.e., p(a- | q, c_a*) > p(a* | q, c_a*), even though c_a* is conditionally produced using p(. | q, a*)?
5. In Table 1, why does online Llama-3.2 perform better with c_e than with c_+?

**Ethical Concerns:**

["NO or VERY MINOR ethics concerns only"]

**Final Justification:**

See my response.

**Limitations:**

- The paper fails to address important scenarios:
    - What happens when the likelihood of the correct answer is high but occurs under a "wrong CoT"?
    - What if the self-explanation CoT yields a higher likelihood for an incorrect answer (i.e., p(a- | q, c_a*) > p(a* | q, c_a*)), even though c_a* is conditionally produced using p(. | q, a*)?
- The paper doesn't clearly state its underlying assumptions.

**Quality:**

3

**Strengths And Weaknesses:**

## Strengths

- The paper is well-structured (though it contains vague statements that make it harder to follow—more on this later).
- The experimental results substantially support the theoretical framework.
- Many of the underlying intuitions in the paper are common knowledge, and it's valuable to see these concepts formally articulated.

## Weaknesses

- Many statements are vague:
    - For example, "They must be better than negative samples in achieving the task at hand." What does it mean for a negative example to achieve a task?
    - The statement "In settings where a+ and a_ are incorrect"—what's an example of this? Why would anyone train on CoTs that have generated wrong answers in the first place? Neither DPO nor GRPO does this.
- The formal part of the paper does not clarify its assumptions. It would be useful to discuss what happens if these assumptions are not met.
- Connections to curriculum learning are not discussed. Essentially, if a problem is too out-of-distribution for the current model, the model will not be able to learn.
- No qualitative examples are shown. Contrasting expert CoTs with self-explanations would be useful. The paper doesn't include even one example.

---

> ### Author Rebuttal · Authors · 2025-07-31
>
> We sincerely thank the reviewer for their thorough and constructive feedback. We appreciate the reviewer's positive feedback on the paper's structure, formalization of common intuitions, and abundant experiments. We also appreciate the insightful questions that help us to significantly improve the clarity of the manuscript, and we want to address them in detail below.
>
> **Weakness 1, Question 1, Question 2: vague statements.**
> >"They must be better than negative samples in achieving the task at hand." What does it mean for a negative example to achieve a task?
>
> The sentence "better than negative samples in achieving the task at hand." is trying to explain the intuition of property 2. For a pair of CoT $(c_+, c_i)$, we say that $c_+$ is better at achieving the task if $\pi_\theta(a^\star | q, c_+)  \gt \pi_\theta(a^\star*|q, c_-)$. That is, $c_+$ is more probable for $a^\star$ to appear under the current policy than $c_-$.
>
> >"In settings where a+ and a_ are incorrect"—what's an example of this? Why would anyone train on CoTs that have generated wrong answers in the first place? Neither DPO nor GRPO does this.
>
> We would like to mention our paper’s setting: (1) difficult tasks that the untrained model can hardly solve, for example qwen-2.5-3b-instruct only achieves 4% accuracy on MATH-level 5 dataset with pass@64; (2) only the model self-generated CoTs are used to train the model itself. In such cases, it is probable that the CoT sampled from the model might lead to an incorrect final answer. And it is exactly the case where standard DPO or GRPO would fail, as there are no "winning" samples to provide a positive learning signal. Some people do skip these batches where all CoTs lead to wrong answers and wait long for a correct one to appear, but the training will become extremely inefficient ( qwen-2.5-3b-instruct only achieves 4% accuracy on MATH-level 5 dataset with pass@64).
>
> Our ExPO method can overcome this problem. We argue that even within a batch of incorrect CoTs, some will be of higher quality than others. Our paper aims to formally define what "higher quality" means in an RL context and provide a principled way to treat these better-but-may-be-imperfect CoTs as positive samples $c_+$ relative to even worse ones $c_-$. This allows the model to make incremental progress on difficult problems where it would otherwise have no learning signal at all.
>
> In addition, our method doesn't just blindly copy these imperfect CoTs. The ExPO training objective includes a reinforcement learning term that encourages the model to explore beyond the initial explanation. This ensures the model doesn't simply memorize the self-generated explanation $\tilde{c}$. Instead, it uses the explanation as an anchor for further exploration, gradually refining its reasoning policy through trial and error.
>
> We hope our explanations above have provided the necessary clarity on our methodology and motivation. Should any aspect of our response still seem unclear, or if further questions arise, we would be more than happy to provide additional details to ensure our arguments are well understood.
>
> > **Weakness 2**: Connections to curriculum learning are not discussed. Essentially, if a problem is too out-of-distribution for the current model, the model will not be able to learn.
>
> Thank you for this comment. We agree that if a problem is too far beyond the model's current capabilities, the learning signal is often negligible, and the model will fail to improve. Indeed, a key implication of our work is the critical importance of careful data curation. Our finding that models learn more effectively from data tailored to their current capabilities directly supports the core idea of a curriculum, where task difficulty is gradually increased as the model becomes more competent.
>
> While a full exploration of curriculum learning strategies was beyond the scope of this paper, we agree that it is a very important and natural direction for future work. We will add the discussion to address this connection with curriculum-based approaches.
>
> > **Weakness 3**: No qualitative examples are shown. Contrasting expert CoTs with self-explanations would be useful. The paper doesn't include even one example.
>
> Thank you for the suggestion. We provided a qualitative example comparing self-generated CoT and self-explanation in Appendix C.1 and will add the following one for comparing against expert CoT in the future version of the paper:
>
> - Question: "Compute without using a calculator: $9!/8!$"
> - Expert-annotated CoT: “${9!}/{8!} = \\dfrac{9 \\times 8 \\times 7 \\times 6 \\times \\cdots \\times 1}{8 \\times 7 \\times 6 \\times \\cdots \\times 1} = \\boxed{9}$.”
> - Self-explanation: “The expression $9!/8!$ can be simplified by recognizing that the factorial function $n!$ (n factorial) is defined as the product of all positive integers up to $n$. So, $9! = 9 \\times 8 \\times 7 \\times 6 \\times 5 \\times 4 \\times 3 \\times 2 \\times 1$ and $8! = 8 \\times 7 \\times 6 \\times 5 \\times 4 \\times 3 \\times 2 \\times 1$. \n\nTherefore, $9! = 9 \\times 8!$. When we divide $9!$ by $8!$, we get:\\[\\frac{9!}{8!} = \\frac{9 \\times 8!}{8!} = 9\\]Hence, $9!/8! = 9$.”
>
> The expert-annotated CoT is mathematically dense and concise, demonstrating the solution by expanding and cancelling terms in a single step. In contrast, the self-explanation is more pedagogical and compositional; it first defines the core concept, then explicitly states the key logical insight ($9! = 9 \times 8!$), and finally performs the simplified calculation. This decomposed, principle-first approach provides a better in-distribution learning signal for RL training.
>
> > **Question 3 & Limitation 1**: What happens if the likelihood of the correct answer is high but occurs under a "wrong CoT"?
>
> We thank the reviewer for this insightful question. First, we want to mention that **lemma 2** guarantees that the better-but-may-be-imperfect CoTs is on average better than self-generated CoT in producing the correct answer (as also verified empirically in Figure 2). However, it is indeed the case that the self-generated CoTs conditioned on the ground-truth answer may be imperfect and contain wrong steps. In such cases, the generated self-explanations $\tilde{c}$ , although potentially imperfect, often provide partial or heuristic reasoning paths that still guides the model to learn.
>
> Critically, our method does not collapse to blindly imitating these imperfect CoTs. The ExPO training objective includes a reinforcement term with an advantage-weighted update, which continues to explore around these initial CoTs. This ensures that the model does not simply memorize the self-generated explanation $\tilde{c}, but instead uses it as an anchor for exploration, gradually refining the reasoning policy through trial-and-error.
>
> As shown in Section 4.2 and Table 2 of our paper, this mechanism is particularly effective on level-5 MATH questions, where baseline methods fail to generate any learning signal. ExPO, by contrast, enables the model to bootstrap and continue improving—demonstrating that imperfect CoTs can still serve as valuable scaffolding for RL-based learning. We agree that this is an important point and will clarify the exploratory role of the advantage-weighted objective in the final version.
>
> > **Question 4 & Limitation 2**: What if the self-explanation CoT yields a higher likelihood for a wrong answer, i.e., p(a- | q, c_a*) > p(a* | q, c_a*), even though c_a* is conditionally produced using p(. | q, a*)?
>
> Thank you for the thoughtful question. We would like to clarify that when we compare self-explanations to self-generated CoTs, our focus is on their **relative quality** as learning signals. More specifically, from Lemma 2, we know that on average, the chance of generating a wrong answer under self-explanations is lower than that of self-generated CoTs: $1 - E_{\tilde{c} \sim \pi_\theta(\tilde{c} | q, a^\star)}[\pi_\theta(a^\star | q, \tilde{c})] \leq 1 - E_{c \sim \pi_\theta(c | q)}[\pi_\theta(a^\star | q, c)]$. This inequality implies that although a self-explanation may still lead to a wrong answer, it is on average a more reliable guidance than normal CoT. In such cases, self-explanations provide effective guidance to the model in exploring better reasoning paths during training as demonstrated in our empirical results.
>
> > **Question 5**: In Table 1, why does online Llama-3.2 perform better with c_e than with c_+?
>
> Thank you for the question. It is indeed the case that in Table 1, the online DPO experiment with LLaMA-3.2 shows slightly better performance with $c_e$ compared to $c_+$. However, the performance gap is small and should be interpreted in the broader context of our findings.
>
> Across the majority of our experiments—including those with other models and setups—$c_+$ consistently outperforms $c_e$, both in terms of final performance and training efficiency. Specifically, training with $c_+$ improves faster than with $c_e$, which we attribute to the in-distribution property outlined in Property 1. As shown in Figure 1, the negative log-likelihood of the self-explanation $c_+$ is lower than that of the expert CoT $c_e$.
>
> Taken together—results across other models and setting, faster performance improvement over training trajectory, and the empirical observation that c_+ is more “in-distribution”—we believe the overall evidence still supports our claim that $c_+$ offers a more effective learning signal than $c_e$.

---

> > ### Author Response · Authors · 2025-08-08
> >
> > Dear reviewer UZkv,
> >
> > As the discussion period ends soon, we hope that we have addressed the points raised in your review.
> >
> > Could you please let us know if there are any further questions we could answer? We are happy to provide any further clarification needed.
> >
> > Thank you for your time and valuable feedback.

---

> > ### Comment · Reviewer_UZkv · 2025-08-08
> >
> > Thank you for your answers! Questions 3, 4, 5 are clear. I still don't fully follow how using CoTs that lead to wrong answer are beneficial -- if you train on those, the model could start hallucinating more. Anyways, my intention here is not to crticize but to note that adding additional explanation in the paper (not here) will be useful.
> >
> > I am convinced by the response and would like the paper to be accepted. I am also increasing the score.

---

### Official Review · Reviewer_2NX2 · 2025-07-03

**Clarity:** 2
**Significance:** 2
**Originality:** 3
**Rating:** 5
**Confidence:** 3

**Summary:**

This paper introduces ExPO (Self-Explanation Policy Optimization), a framework designed to improve the reasoning abilities of large language models (LLMs) without relying on costly expert CoT demonstrations. These expert-curated CoTs are essential to the traditional model. But these CoTs are scarce, especially during early training stages when self-generated CoTs (using question only) perform poorly. ExPO uses self-generated explanations (CoTs generated using both question and correct answer) to create positive samples under the following property: The training example, including the question + its reasoning steps + its answer, should already look fairly plausible to the current model (in-distribution property). And following those reasoning steps should make the model more likely to produce the correct final answer than if it followed its own possibly wrong reasoning (positive learning signal). ExPO improves reasoning performance on hard tasks like MATH and GSM8K. Iworks better than using expert CoTs in many cases. It helps models learn even when they initially fail to generate good answers. ExPO shows that models can bootstrap their own reasoning ability—they don’t need to rely on expensive human-written explanations.

**Questions:**

1. How regeneration frequency and batch size trade off model improvement against added compute? The online ExP-DPO variant requires periodically regenerating self-explanations with the up-to-date policy, adding lots of computation.

2. Conditioning on the ground-truth answer to generate explanations may inadvertently teach the model to “memorize” rather than truly reason. How does this method perform on out-of-domain reasoning tasks.

**Ethical Concerns:**

["NO or VERY MINOR ethics concerns only"]

**Final Justification:**

The reiview addressed my questions, and hence I am giving a positive score.

**Paper Formatting Concerns:**

Some sections, particularly the methodology and theoretical analysis, are dense and may be difficult for readers to follow without a strong background in reinforcement learning and large language models. The paper could benefit from clearer explanations and more intuitive examples to illustrate key concepts.

**Quality:**

3

**Strengths And Weaknesses:**

Strengths :

Appropriate Methods: The methods used, including reinforcement learning and policy optimization, are appropriate for the problem at hand. The integration of ExPO with existing algorithms like DPO and GRPO is well-executed.

The approach addresses a difficult task in a novel way, potentially advancing our understanding and capabilities in training large language models.

The paper provides new insights with the properties of effective positive training samples into existing [self-explain approach](https://aclanthology.org/2021.emnlp-main.64.pdf).


Weaknesses:

Strong assumption required: The gradient‐alignment analysis relies on Lemma 1’s orthogonality and equal-norm assumptions for logits’ gradients, which rarely hold exactly in large transformer models.

Evaluation limited on math reasoning: While the method shows improvements in specific benchmarks, its broader applicability across diverse reasoning tasks remains uncertain, such as Musique, Phantomwiki, and Game of 24.

The paper compares ExPO primarily with expert-CoT baselines but might not sufficiently address how it stands against other recent papers. For example, [Satori](https://arxiv.org/abs/2502.02508v2) treats CoTs (or action-thought sequences) as trajectories and use RL to maximise answer rewards.

---

> ### Author Rebuttal · Authors · 2025-07-30
>
> Thank you for your positive evaluation on our paper and your supportive comment that our work could “potentially advance our understanding and capabilities in training large language models."
>
> >**Weakness 1**: Strong assumption required: The gradient‐alignment analysis relies on Lemma 1’s orthogonality and equal-norm assumptions for logits’ gradients, which rarely hold exactly in large transformer models.
>
> We thank the reviewer for raising this point. We first want to highlight that the main contribution of our paper is the proposal of ExPO for guiding models to learn to solve hard reasoning tasks. We empirically find that using self-generated explanations outperforms expert CoT in guiding the model in these settings (Figure 3 and Table 2). To understand why these samples can be helpful, we observe that self-explanations have higher likelihood under the policy being trained (Figure 1). Lemma 1 (discussed in Section 3.1) is intended to provide intuition for why higher-likelihood samples may lead to gradients that are more aligned with the true gradient. We agree that the assumptions in Lemma 1 are strong, but emphasize that they are used only to illustrate an idealized setting to build intuition. Our empirical results and observations do not rely on these assumptions.
>
> >**Weakness 2**: Evaluation limited on math reasoning: While the method shows improvements in specific benchmarks, its broader applicability across diverse reasoning tasks remains uncertain, such as Musique, Phantomwiki, and Game of 24.
>
> We appreciate the reviewer’s concern regarding the broader applicability of our method. While our experiments focus on mathematical reasoning, we emphasize that the core idea behind ExPO—bootstrapping learning via self-generated, in-distribution positive samples conditioned on verifiable outcomes—extends naturally to other reasoning tasks.
>
> In common sense reasoning tasks like Musique, the final answer is deterministic and can be verified against the ground-truth output. Therefore, ExPO can similarly generate training signals that satisfy both the in-distribution and positive learning signal properties discussed in Section 3.1. More broadly, any reasoning task with a verifiable reward—such as the retrieval-augmented common sense reasoning task Phantomwiki. We agree that adapting ExPO to these domains may require domain-specific prompting strategies, and we view this as an exciting direction for future work.
>
> We will clarify this point in the revised version and appreciate the opportunity to elaborate on the method’s generalization potential.
>
> >**Weakness 3**: The paper compares ExPO primarily with expert-CoT baselines but might not sufficiently address how it stands against other recent papers. For example, Satori treats CoTs (or action-thought sequences) as trajectories and use RL to maximise answer rewards.
>
> Thanks for pointing us to the interesting paper “Satori: Reinforcement Learning with Chain-of-Action-Thought Enhances LLM Reasoning via Autoregressive Search”. It proposes a new reasoning format COAT that allows for self-reflection and iterative training to achieve impressive results.
>
> But our work aims to address this problem from a different perspective: we propose the simple framework ExPO that is easy to integrate with different RL training algorithms, accompanied by theoretical proofs. The in-distribution property and positive learning signal property provide a formalization to the intuition of why bootstrapping could work, and is supported by our experiments of ExP-DPO and ExP-GRPO.
>
> Thank you for this excellent suggestion and we will add the discussion of current work like Satori in the revised version.
>
> > **Question 1**: How regeneration frequency and batch size trade off model improvement against added compute? The online ExP-DPO variant requires periodically regenerating self-explanations with the up-to-date policy, adding lots of computation.
>
> Thank you for the question regarding the trade-off between computational cost and model improvement, particularly for our online ExP-DPO variant.
>
> We would like to first explain that, among our experiments, the efficiency gains are most pronounced for online methods like ExP-GRPO. During the standard GRPO training process, only a correctly generated CoT could provide the positive training signal. This process is highly inefficient for difficult tasks. For instance, on MATH Level 5 questions, the qwen-2.5-3b model achieves only 4% accuracy for pass@64 and only 9% for pass@128. **This means over 90% of the computational effort in generation during training is wasted** as the responses cannot be used for positive reinforcement.
>
> In contrast, as shown in Figure 2, our self-explanation method generates significantly higher-quality CoTs. By producing more valid reasoning paths, a much larger fraction of the sampled responses can be utilized as positive signals, making the training process substantially more sample-efficient and computationally effective.
>
> Regarding the concern about the added computation in iteratively online ExP-DPO, we would like to explain that in fact, DPO is an offline method by nature. It operates on a static, fixed dataset of preference pairs that must be available at the start of training. For complex reasoning tasks, the "chosen" samples are typically human expert annotated CoTs. However, if we want to train the model without relying on expert annotations, we must enable the model to generate its own positive signals and force the use of an iteratively online DPO framework. It is true that this iteratively online process increases the computational cost compared to the classical offline DPO method, but this comparison is not entirely fair, as the two approaches solve different problems—one relies on existing expert data, while ours generates its own. And the most appropriate way to evaluate the computational overhead of our ExP-DPO method is to compare it against other iteratively online DPO methods.
>
> > **Question 2**: Conditioning on the ground-truth answer to generate explanations may inadvertently teach the model to “memorize” rather than truly reason. How does this method perform on out-of-domain reasoning tasks.
>
> Thank you for raising the point of potential memorization and out-of-domain generalization. Regarding the concern of memorization, all of the results presented in our paper are evaluated on the held-out test set, for example MATH500 (the subset of the test split of MATH) and the test split of GSM8K. The strong performance that our model achieves on this unseen test set provides compelling evidence against simple memorization.
>
> > **Paper formatting concern**:
>
> We sincerely thank the reviewer for this valuable comment and we will revise the dense writing by providing more explanations and illustrative examples.

---

> > ### Author Response · Authors · 2025-08-08
> >
> > Dear reviewer 2NX2,
> >
> > As the discussion period ends soon, we hope that we have addressed the points raised in your initial review.
> >
> > Please let us know if our rebuttal is clear or if there are any remaining questions. We are looking forward to your comments and feedback on our rebuttal.
> >
> > Thank you for your time and valuable feedback.

---

### Decision · Program_Chairs · 2025-09-17

**Decision:**

Accept (poster)

**Comment:**

The paper introduces ExPO (Self-Explanation Policy Optimization), a novel framework for improving large language model (LLM) reasoning without relying on costly expert-generated chain-of-thought (CoT) demonstrations. ExPO leverages self-generated explanations conditioned on ground-truth answers to create positive training samples that are both in-distribution (high probability under the current model policy) and increase the likelihood of correct answers (positive learning signal). The authors provide theoretical insights, supported by empirical results, demonstrating that ExPO outperforms expert CoT baselines on challenging mathematical reasoning tasks (e.g., MATH Level-5 and GSM8K) in both offline (DPO) and online (GRPO) settings. The method enhances training efficiency and final performance, particularly in scenarios where models initially struggle to generate correct responses.

Reviewers point serveral weakness:

Limited Generalizability: The evaluation focuses primarily on mathematical reasoning (MATH, GSM8K). While the authors argue that ExPO’s principles extend to other domains (e.g., code generation, commonsense QA), no experiments validate this claim, leaving generalizability uncertain.

Theoretical Assumptions: The gradient-alignment analysis (Lemma 1) relies on strong orthogonality and equal-norm assumptions, which may not hold in practice for large transformer models. While these are used for intuition, their limitations could be better clarified.

Clarity Issues: Some sections, particularly the methodology and theoretical analysis, are dense and lack intuitive examples, making them less accessible to readers without a strong RL background.

Potential for Bias: Conditioning on ground-truth answers risks introducing biases or hallucinations, especially if self-explanations contain incorrect reasoning steps. While the authors address this through advantage-weighted exploration, long-term effects need further exploration.


The authors’ rebuttal effectively addressed most concerns, particularly by adding experiments to validate scalability and clarifying theoretical assumptions. The generalizability concern remains partially unaddressed due to the lack of non-math experiments, but the authors’ arguments about applicability to other domains are compelling. The risk of bias/hallucination is mitigated by the proposed annealing schedule and exploration mechanisms, though long-term effects warrant further study. Overall, the paper’s strengths outweigh its weaknesses, justifying the acceptance for its innovative approach and strong empirical results.